# PC-Net: Weakly Supervised Compositional Moment Retrieval via Proposal-Centric Network

Mingyao Zhou[1,2,3]     Hao Sun[1,2,3*]     Wei Xie[1,2,3]     Ming Dong[1,2,3]
Chengji Wang[1,2,3]     Mang Ye[4]

[1]School of Computer Science, Central China Normal University
[2]Hubei Provincial Key Laboratory of Artificial Intelligence and Smart Learning,
Central China Normal University
[3]National Language Resources Monitoring and Research Center for Network Media,
Central China Normal University
[4]School of Computer Science, Wuhan University

## Abstract

With the exponential growth of video content, aiming at localizing relevant video moments based on natural language queries, video moment retrieval (VMR) has gained significant attention. Existing weakly supervised VMR methods focus on designing various feature modeling and modal interaction modules to alleviate the reliance on precise temporal annotations. However, these methods have poor generalization capabilities on compositional queries with novel syntactic structures or vocabulary in real-world scenarios. To this end, we propose a new task: weakly supervised compositional moment retrieval (WSCMR). This task trains models using only video-query pairs without precise temporal annotations, while enabling generalization to complex compositional queries. Furthermore, a proposal-centric network (PC-Net) is proposed to tackle this challenging task. First, video and query features are extracted through frozen feature extractors, followed by modality interaction to obtain multimodal features. Second, to handle compositional queries with explicit temporal associations, a dual-granularity proposal generator decodes multimodal global and frame-level features to obtain query-relevant proposal boundaries with fine-grained temporal perception. Third, to improve the discrimination of proposal features, a proposal feature aggregator is constructed to conduct semantic alignment of frames and queries, and employ a learnable peak-aware Gaussian distributor to fit the frame weights within the proposals to derive proposal features from the video frame features. Finally, the proposal quality is assessed based on the results of reconstructing the masked query using the obtained proposal features. To further enhance the model's ability to capture semantic associations between proposals and queries, a quality margin regularizer is constructed to dynamically stratify proposals into high and low query-relevance subsets and enhance the association between queries and common elements within proposals, and suppress spurious correlations via inter-subset contrastive learning. Notably, PC-Net achieves superior performance with 54% fewer parameters than prior works by parameter-efficient design. Experiments on Charades-CG and ActivityNet-CG demonstrate PC-Net's ability to generalize across diverse compositional queries. Code is available at `https://github.com/mingyao1120/PC-Net`.

## 1 Introduction

Video moment retrieval aims to identify the precise timestamps in videos corresponding to user queries, which has gained significant research attention. Fully supervised methods heavily rely on

---

*Corresponding author: `haosun@ccnu.edu.cn`

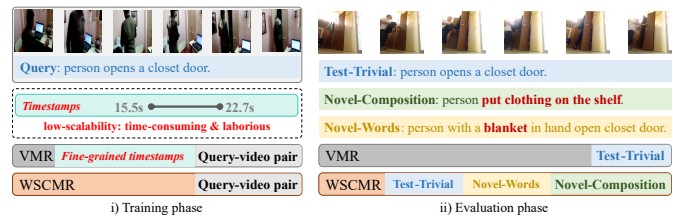
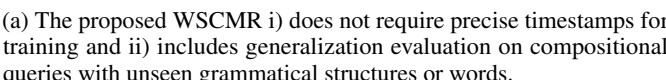
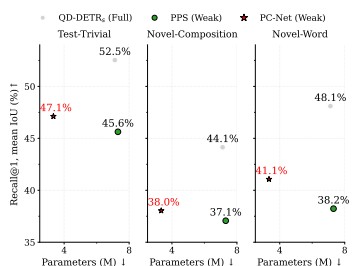

(a) The proposed WSCMR i) does not require precise timestamps for training and ii) includes generalization evaluation on compositional queries with unseen grammatical structures or words.

(b) Performance comparison of the proposed method with the SOTAs.

Figure 1: Comparison of the proposed method with existing methods on paradigm and performance. Here, QD-DETR$_s$ [1] is the fully supervised method, and PPS [2] is the weakly supervised method.

datasets with labor-intensive precise timestamps, limiting their scalability in practical applications [3, 4, 5], as illustrated in Figure 1(a). Although weakly supervised methods [2, 6, 7] alleviate the annotations burden by training solely on video-query pairs, yet they fail to address a core practical challenge: generalization to queries with novel syntactic structures or unseen vocabulary [8, 9].

Recently, the compositional moment retrieval task has focused on the generalization of VMR methods across various query styles in real-world scenarios, utilizing compositional queries with novel syntactic structures and vocabulary to evaluate the generalization ability of models [1, 8, 9]. However, this task is still fully supervised and relies on massive, precise timestamps. To this end, we propose a new task, named Weakly Supervised Compositional Moment Retrieval (WSCMR), which aims to train models without requiring timestamps and enables them to generalize across compositional queries. This task is fundamentally challenging due to weak pairwise supervision and limited query style in training, and the need to generalize to compositional queries with diverse temporal and semantic dependencies.

An intuitive approach to address the proposed WSCMR is to leverage existing weakly supervised models [7, 10]. However, their inherent limitations hinder them from effectively handling compositional queries. Specifically, these models typically follow a three-stage process around the proposal: boundary generation, feature aggregation, and quality assessment. First, existing methods only generate proposal boundaries through global video-query associations, which lack fine-grained temporal perception and are difficult to handle compositional queries with explicit temporal logic such as "the second person turns on the light" [11, 12, 13]. In feature aggregation, existing methods employ a fixed Gaussian distribution based on the proposal boundaries to aggregate proposal features from video frame features. These methods ignore the semantic gap between frame and query features [14, 15], and the variability in action durations across different query compositions [8, 9], resulting in low discrimination of proposal features. Furthermore, after the quality assessment based on autoregressively reconstructing the masked query with proposal features, existing methods [2, 6, 7, 10] construct negative samples only from the proposal with the lowest reconstruction loss. This discards partially relevant proposals, preventing the model from learning to associate visual representations with specific query elements and undermining compositional generalization [1].

To address these challenges, a proposal-centric network (PC-Net) is constructed. First, the video and query features are extracted through frozen feature extractors, and the modal interaction based on attention mechanism is then conducted to obtain multimodal features consistent with prior works [2, 7]. Secondly, to capture explicit temporal associations in compositional queries and enhance semantic consistency, the dual-granularity proposal generator decodes frame-level and global multimodal features to produce relevant boundaries with fine-grained temporal perception. Thirdly, to alleviate the semantic gap between the frames and the query and difficulty in modeling the variability of action duration of the fixed Gaussian distribution, the proposal feature aggregator first constructs feature triplets (queries, relevant frames, and irrelevant frames) to map them to a unified semantic space, and then dynamically adjusts the Gaussian peak region for ensuring discriminative proposal features that align with compositional query semantics. Finally, to assist the model in capturing the semantic associations between visual representations and query elements, a quality margin regularizer is constructed. The regularizer divides the proposals into subsets with high and low query relevance,

then helps the model capture the query-related visual elements that co-occur in the proposals and suppresses potential spurious associations through contrastive learning between subsets, indirectly optimizing the proposal representation to cope with potential compositional queries.

PC-Net uses a generator with fewer parameters and efficient constraint losses to optimize feature aggregation and quality assessment around proposals. Experiments show a 54% parameter reduction while outperforming baselines in novel query generalization, demonstrating high efficiency and strong generalization. In summary, the contributions of this paper are as follows:

- The Weakly Supervised Compositional Moment Retrieval (WSCMR) task is introduced, aiming to train a moment retrieval model capable of generalizing to queries with novel syntactic structures and vocabulary while eliminating the need for precise timestamps. Furthermore, the limitations of existing methods are analyzed and validated.

- PC-Net is proposed, which employs a compact proposal generator, complemented by an efficient aggregator and regularizer to guide proposal feature aggregation and capture semantic associations for enhancing the model's ability of compositional generalization.

- The framework achieves superior performance using 54% fewer parameters than comparable methods, demonstrating the efficiency of the design and the generalizability of the proposed framework for novel queries.

## 2 Related Works

### 2.1 Video Moment Retrieval

Video moment retrieval aims to locate specific temporal segments in videos that semantically match textual queries. This field has evolved through two primary paradigms: fully supervised and weakly supervised approaches. Fully supervised methods, pioneered by Gao et al. [16], employ either proposal-based [16, 17, 18, 19] or regression-based [20, 21, 22, 23, 24] strategies. Proposal-based methods generate candidate segments using sliding windows [16] or learnable networks [18], followed by ranking based on query relevance. However, these methods face computational inefficiency due to dense proposal sampling and sensitivity to proposal quality [4]. Regression-based alternatives [20, 23] directly predict temporal boundaries by fusing multimodal features, enabling end-to-end training but struggling with compositional queries due to rigid annotation patterns [9]. Recent advances improve compositional generalization through decompose-and-reconstruct strategies [8] and saliency-aware contrastive learning with large language models [1]. Despite progress, fully supervised methods remain constrained by their reliance on precise temporal annotations, which are labor-intensive and prone to subjective inconsistencies [25, 26], weakening their scalability.

Weakly supervised methods eliminate temporal annotations by learning from relevant video-query pairs. Multiple instance learning (MIL)-based approaches [27, 28, 29, 30] optimize global video-query alignment but often miss fine-grained correspondences. Masked query reconstruction methods [6, 10, 31, 32] enhance semantic alignment through cross-modal reconstruction. SCN [33] integrates masked reconstruction and contrastive losses, while CNM [10] introduces Gaussian masks and triplet learning to refine proposals. Subsequent innovations include CPL [7] with learnable Gaussian distributions, counterfactual reasoning extensions [32], and mixed Gaussian frameworks [2]. Methods like PPS [34] and QMN [35] further explore iterative refinement but remain limited by fixed feature aggregation and simple contrastive objectives. Notably, weakly supervised methods ignore the generalization requirements of diverse query styles in practical applications, making it difficult to cope with queries with novel grammatical structures or words. In contrast, the proposed WSCMR task not only abandons the need for accurate timestamps but also requires the model to have good generalization capabilities for novel queries, improving practical usability.

### 2.2 Multi-modal Semantic Alignment

To address the inherent heterogeneity of multimodal data across temporal, spatial, and semantic dimensions [36, 37], modality alignment has evolved from shallow feature matching to deep joint feature modeling. Early approaches, such as projecting inputs into a shared latent space [38], reduced intra-class variation while enhancing inter-class discrimination. Recent advances leverage semantic-level alignment [39] and contrastive learning objectives [40, 41, 42] to model interactions. For

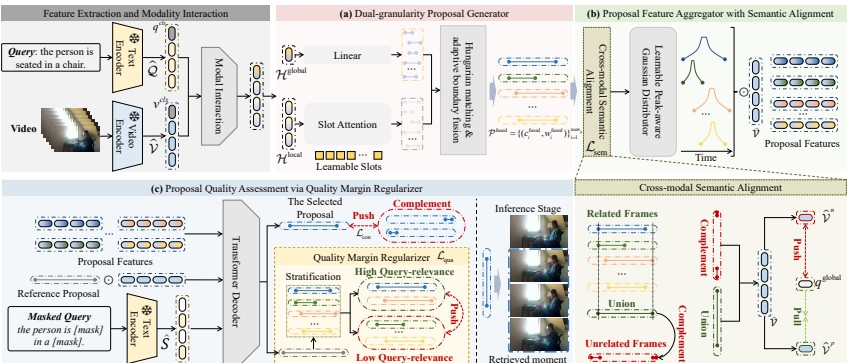

Figure 2: The overall framework of the proposed PC-Net for solving the WSCMR task consists of a dual-granularity proposal generator that uses the dual-granularity association of video-query pairs to obtain query-relevant boundaries with frame-level temporal perception, a proposal feature aggregator for modeling discriminative features, and a query margin regularizer that enables the model to capture visual elements relevant to the query and suppresses potential spurious correlations.

instance, cycle-consistency loss aligns video segments with query words [36], while multi-level contrastive objectives capture hierarchical video-query relationships for improved retrieval [37]. Cheng et al. [1] further refine this paradigm by generating curriculum-based negative queries via large language models. Considering that the video contains a large number of frames that are irrelevant to the query and relevant timestamps are not available, the alignment of the proposed method is not directly aligned at the video level, but based on the proposal frames related to the query, to perform feature alignment, which facilitates reliable proposal quality assessment.

## 3 Methods

### 3.1 Feature Extraction and Modality Interaction

Consistent with existing weakly supervised methods [2, 7, 10], GLoVe [43] is used to extract query features, and I3D [44] and C3D [45] are used to extract Charades-CG [16] and ActivityNet-CG [46] features, respectively. The word-level query features are represented as: $\mathcal{Q} = [q_1, q_2, \ldots, q_N] \in \mathbb{R}^{N \times d_q}$, where $N$ is the number of words and $d_q$ is the hidden dimension of the query words. The frame-wise video features are represented as: $\mathcal{V} = [v_1, v_2, \ldots, v_T] \in \mathbb{R}^{T \times d_v}$, where $T$ is the number of video frames and $d_v$ is the hidden dimension of the video frames. $\mathcal{Q}$ and $\mathcal{V}$ will be mapped to the same latent dimension $d$ to facilitate subsequent proposal generation, namely $\hat{\mathcal{Q}} = \mathrm{Linear}(\mathcal{Q}) \in \mathbb{R}^{N \times d}, \hat{\mathcal{V}} = \mathrm{Linear}(\mathcal{V}) \in \mathbb{R}^{T \times d}$. Among them, $\mathrm{Linear}(\cdot)$ is a linear layer. To fully explore the semantics of a given query and cope with potentially novel queries, an attention mechanism is used to fuse the query's word-level features $\hat{\mathcal{Q}}$ with the video's frame-level features $\hat{\mathcal{V}}$, resulting in a multimodal features $\mathcal{H}^{\mathrm{global}} \in \mathbb{R}^{1 \times d}$ with the overall semantics understanding and $\mathcal{H}^{\mathrm{local}} \in \mathbb{R}^{T \times d}$ with fine-grained temporal information. More details can be found in Appendix A.1.

### 3.2 Dual-granularity Proposal Generator

Existing weakly supervised methods [2, 7, 10, 32] rely solely on global multimodal associations to generate proposals. This approach inherently lacks fine-grained temporal perception and ignores frame-level query relevance, resulting in imprecise boundaries with blurred temporal logic. Consequently, these methods exhibit limited generalization to queries involving explicit temporal dependencies [11, 12]. To address this limitation, the dual-granularity proposal generator is constructed to capture global semantic consistency and local temporal precision jointly, for generating boundaries with both holistic scene understanding and fine-grained temporal awareness.

First, the global multimodal feature $\mathcal{H}^{\mathrm{global}}$ is linearly mapped to global proposals: $\mathcal{P}^{\mathrm{global}} = \mathrm{Linear}(\mathcal{H}^{\mathrm{global}}) \in \mathbb{R}^{num_p \times 2}$. $\mathcal{P}^{\mathrm{global}} = \{(c_i^{\mathrm{global}}, w_i^{\mathrm{global}})\}_{i=1}^{num_p} \in [0, 1]^{num_p \times 2}$ contains proposals with global semantic coherence. Here, $c_i$ and $w_i$ respectively refer to the center point and duration of the $i$-th proposal; the same notation applies hereafter. Second, to incorporate frame-level temporal details, frame-wise multimodal features $\mathcal{H}^{\mathrm{local}}$ are processed using a slot attention mechanism [47].

$num_p$ all-zero learnable proposal slots $\mathcal{P}_0^{\text{local}} \in \mathbb{R}^{num_p \times d}$ are initialized and are iteratively refined via $K$ rounds of attention interactions with $\mathcal{H}^{\text{local}}$. The update rule at the $k$-th iteration is defined as:

$$\mathcal{P}_k^{\text{local}} = \text{Softmax}\left(\frac{(\mathcal{H}^{\text{local}})(\mathcal{P}_{k-1}^{\text{local}})^{\mathsf{T}}}{\sqrt{d}}\right) \cdot \mathcal{H}^{\text{local}} + \mathcal{P}_{k-1}^{\text{local}}. \tag{1}$$

After $K$ iterations, local proposals $\mathcal{P}_K^{\text{local}} = \{(c_i^{\text{local}}, w_i^{\text{local}})\}_{i=1}^{num_p} \in [0,1]^{num_p \times 2}$ are generated, which involve explicit temporal boundaries and query-related frame semantics.

To combine the strengths of global and local proposals, the Hungarian matching [48] is employed to adaptively match and fuse their boundaries. This adaptive fusion mechanism ensures that the final proposal set $\mathcal{P}^{\text{fused}} = \{(c_i^{\text{fused}}, w_i^{\text{fused}})\}_{i=1}^{num_p}$ retains both global semantic consistency and fine-grained temporal perception. More details can be found in Appendix A.2.

### 3.3 Proposal Feature Aggregator with Semantic Alignment

Existing weakly supervised methods [7, 32] rely on constructing a fixed Gaussian distribution based on the proposal boundary to aggregate proposal features from frame features. This approach has two key limitations: i) there is a semantic gap between frame features and query features [14, 15]; ii) the fixed distribution is difficult to fit the diversity of action durations across different query combinations, which ultimately leads to a decrease in feature discrimination ability [2]. To address these issues, a proposal feature aggregator with two components (Figure 2(b)) is constructed. First, cross-modal semantic alignment constructs a unified feature space via triplets of queries, related/irrelevant video frames, bridging the semantic gap between queries and frames. Second, a learnable peak-aware Gaussian distributor dynamically adjusts the Gaussian peak region to simulate diverse durations, ensuring discriminative proposal features consistent with the compositional query semantics (Figure 7).

**Cross-modal Semantic Alignment.** To bridge the gap between videos and queries, feature triplets of queries, relevant video segments, and irrelevant video segments are constructed to map them into a unified semantic space based on contrastive learning. First, the query-relevant video feature $\hat{\mathcal{V}}_i^p \in \mathbb{R}^{1 \times d}$ extraction process by a weighted average of the original video features $\hat{\mathcal{V}}$ as follows.

$$\hat{\mathcal{V}}_i^p = \frac{1}{|M_i|} \sum_{t \in M_i} \hat{\mathcal{V}}_t \in \mathbb{R}^{1 \times d}, M_i = \left\{ t \mid \lfloor c_i^{\text{fused}}T \rfloor - \frac{w_i^{\text{fused}}T}{2} \le t < \lfloor c_i^{\text{fused}}T \rfloor + \frac{w_i^{\text{fused}}T}{2} \right\}. \tag{2}$$

Here, $\lfloor \cdot \rfloor$ represents a floor operation and $T$ is the number of frames in the video. $(c_i^{\text{fused}}, w_i^{\text{fused}})$ is the center and width of the $i$-th proposal $\mathcal{P}_i^{\text{fused}}$. $M_i$ is the start and end frame sequence of the $i$-th proposal, and $|M_i|$ is the number of frames in the video feature that fall within the $i$-th proposal. The query-irrelative video feature $\hat{\mathcal{V}}_i^n \in \mathbb{R}^{1 \times d}$ is obtained by a weighted average of the complement of the relevant area or the average of the global video, as follows:

$$\hat{\mathcal{V}}_i^n = \frac{1}{T - |M_i|} \sum_{t \notin M_i} \hat{\mathcal{V}}_t \cdot \mathbf{1}_{\{|M_i| < T\}} + \frac{1}{T} \sum_{t=1}^{T} \hat{\mathcal{V}}_t \cdot \mathbf{1}_{\{|M_i| = T\}}. \tag{3}$$

Here, conditional branch $\mathbf{1}_{\{A\}} = 1$ where condition $A$ is true otherwise 0. The semantic contrastive loss is as follows. Here, $\gamma = 0.5$ is the margin coefficient. $\text{sim}(\cdot)$ is the cosine similarity function.

$$\mathcal{L}_{\text{sem}} = \frac{1}{num_p} \sum_{i=1}^{num_p} \max\left(0, \text{sim}(q^{\text{global}}, \hat{\mathcal{V}}_i^n) - \text{sim}(q^{\text{global}}, \hat{\mathcal{V}}_i^p) + \gamma\right), \tag{4}$$

where $q^{\text{global}}$ contains the global context of the query from modality interaction, see Appendix A.1.

**Learnable Peak-aware Gaussian Distributor.** Following existing works [6, 7, 10], the boundary of the $i$-th proposal is used to generate the corresponding initial fixed Gaussian distribution $G_i \in \mathbb{R}^T$ (Appendix A.3). $G_i$ is the frame-level weight of the $i$-th proposal. However, a fixed Gaussian distribution has difficulty modeling the duration of various action compositions [9]. Therefore, we

propose to adaptively redistribute the peak weights of the Gaussian distribution as follows.

$$M_i(t) = \frac{1}{1 + e^{-1000 \cdot \eta_i(t)}}, \text{ where } \eta_i(t) = \beta \sigma_i - |x_t - c_i^{\text{fused}}|, \tag{5}$$

$$W_i(t) = G_i(t) \cdot (1 - M_i(t)) + M_i(t). \tag{6}$$

Here, $W_i \in \mathbb{R}^T$ denotes the dynamically adjusted frame-level weight of the $i$-th proposal. $\eta_i(t)$ determines whether to retain the original Gaussian value by measuring the distance between each time point $x_t$ and the proposal center $c_i^{\text{fused}}$. Specifically, the closer $x_t$ is to the proposal center (peak area), the larger $\eta_i(t)$ becomes (greater than 0), and thus the value of $M_i(t)$ approaches 1, effectively assigning a weight of 1 to the frame at time $x_t$. Conversely, as $M_i(t)$ approaches 0, the original Gaussian value is preserved. The parameter $\beta$ is a learnable scaling factor that adjusts the width of the peak area to capture diverse action durations across compositional queries. Details of $\sigma_i$ and $x_t$ are provided in Appendix A.3.

### 3.4 Proposal Quality Assessment via Quality Margin Regularizer

To accurately measure the proposal quality and select the best proposals, the proposal quality assessment process follows previous work [7, 10]. The original query is partially masked and then reconstructed from the proposal features aggregated from the original video frame features using the adjusted Gaussian distribution, and the reconstruction loss is used to measure the proposal quality. However, existing methods [2, 6, 7, 32] only construct negative proposals based on the proposal with the smallest reconstruction loss for contrastive learning, discarding partially relevant proposals, which makes it difficult for the model to capture the subtle semantic associations between relevant proposals and queries. Therefore, the quality margin regularizer is built to use the reconstruction quality of the reference proposal (i.e.,whole video frames) to dynamically divide the proposal set into high and low query-relevance subsets, and correlations of co-occurring visual elements related to the query are established in proposals through inter-subset contrastive learning.

**Proposal Quality Assessment.** Consistent with previous methods [2, 7, 10], the reference and negative proposals are generated to constrain the contrast of the best proposals. The weight of the referenced proposal is a vector of all ones of length $N$, and the reconstruction loss through Formula (18) is $\mathcal{L}_r^{re}$, more details can be found in Appendix A.4. Suppose the proposal with the smallest reconstruction loss is denoted as $\mathcal{P}_O^{\text{fused}}$, then the corresponding center point and duration are $(c_O^{\text{fused}}, w_O^{\text{fused}})$ respectively. The reconstruction loss of $\mathcal{P}_O^{\text{fused}}$ through Formula (18) is $\mathcal{L}_O^{re}$. The complement of $(c_O^{\text{fused}}, w_O^{\text{fused}})$ is shown in Figure 2 (c), including left negative proposal $\mathcal{P}_{n_1}^{\text{fused}}$ and right negative proposal $\mathcal{P}_{n_2}^{\text{fused}}$. Through the above proposal feature aggregation and mask query reconstruction, the corresponding two negative proposal reconstruction losses $\mathcal{L}_{n_1}^{re}$ and $\mathcal{L}_{n_2}^{re}$ are obtained. The contrast loss based on the optimal proposal $\mathcal{P}_O^{\text{fused}}$ is as follows. Among them, $\theta_1$ and $\theta_2$ are hyperparameters, and $\theta_1 < \theta_2$.

$$\mathcal{L}_{\text{con}} = \max\left(0, \mathcal{L}_O^{re} - \mathcal{L}_r^{re} + \theta_1\right) + \max\left(0, \mathcal{L}_O^{re} - \mathcal{L}_{n_1}^{re} + \theta_2\right) + \max\left(0, \mathcal{L}_O^{re} - \mathcal{L}_{n_2}^{re} + \theta_2\right), \tag{7}$$

**Quality Margin Regularizer.** Previous relatively simple negative sample contrast makes it difficult for the model to identify subtle semantic differences between query-relevant proposals and avoid spurious associations. Therefore, the quality margin regularizer is built to fully utilize the given query and amplify the feature representation distinctions between high and low query relevance proposals, as shown in Figure 2 (c). First, the original positive proposals $\mathcal{L}^{re} \in \mathbb{R}^{num_p}$ are split into two subsets according to the quality of the reference proposal $\mathcal{L}_r^{re}$, and then extract the average quality of the high and low query-relevance proposal sets for contrast.

$$\mathcal{L}_{\text{high}} = \frac{1}{|\mathcal{X}|} \sum_{i \in \mathcal{X}} \mathcal{L}_i^{re}, \mathcal{X} = \left\{i \mid \mathcal{L}_i^{re} < \mathcal{L}_r^{re}\right\}. \mathcal{L}_{\text{low}} = \frac{1}{|\mathcal{Y}|} \sum_{i \in \mathcal{Y}} \mathcal{L}_i^{re}, \mathcal{Y} = \left\{i \mid \mathcal{L}_i^{re} \geq \mathcal{L}_r^{re}\right\}. \tag{8}$$

Among them, $\mathcal{L}_{\text{high}}$ and $\mathcal{L}_{\text{low}}$ are the average quality of high/low query-relevance proposals, respectively. The final quality contrastive loss between proposals is $\mathcal{L}_{\text{qua}} = \max\left(\mathcal{L}_{\text{high}} - \mathcal{L}_{\text{low}} + \theta_3, 0\right)$. Here, $\theta_3$ is a margin hyperparameter that controls the strictness of the contrast.

### 3.5 Optimization Goals

The loss in this paper consists of five parts: proposal reconstruction loss $\mathcal{L}_O^{re}$, proposal contrast loss $\mathcal{L}_{\text{con}}$, proposal diversity constraint $\mathcal{L}_{\text{div}} = \|WW^\top - \lambda I\|_F^2$ from CPL [7] calculated by proposal frame-level weights $W \in \mathbb{R}^{num_p \times T}$, semantic alignment loss $\mathcal{L}_{\text{sem}}$, and quality contrastive loss $\mathcal{L}_{\text{qua}}$.

$$\mathcal{L}_{\text{total}} = \mathcal{L}_O^{re} + \lambda_{\text{con}}\mathcal{L}_{\text{con}} + \lambda_{\text{div}}\mathcal{L}_{\text{div}} + \lambda_{\text{sem}}\mathcal{L}_{\text{sem}} + \lambda_{\text{qua}}\mathcal{L}_{\text{qua}} \tag{9}$$

Among them, $\lambda_*, * \in \{\text{con}, \text{div}, \text{sem}, \text{qua}\}$ is the coefficient of the corresponding loss. During inference, the proposal with the smallest reconstruction loss $\mathcal{L}_O^{re}$ is taken as the retrieval result.

## 4 Experiments

Table 1: Comparative performance on the Charades-CG dataset. Here, R$n$@$m$ denotes the Recall@$n$ metric under an IoU threshold of $m$. Results in **bold** are optimal, underlined results are suboptimal. "Full Supervision" represents methods that use precise timestamp annotations during training, and "Weak Supervision" represents methods that train only on video-text pairs without timestamp annotations (results are reproduced based on public implementations). Same as below.

| | Method | Params | Test-Trivial | | | Novel-Composition | | | Novel-Word | | |
|---|---|---|---|---|---|---|---|---|---|---|---|
| | | | R1@0.5 | R1@0.7 | mIoU | R1@0.5 | R1@0.7 | mIoU | R1@0.5 | R1@0.7 | mIoU |
| Full Supervision | TMN [49] | - | 18.75 | 8.16 | 19.82 | 8.68 | 4.07 | 10.14 | 9.43 | 4.96 | 11.23 |
| | TSP-PRL [22] | - | 39.86 | 21.07 | 38.41 | 16.30 | 2.04 | 13.52 | 14.83 | 2.61 | 14.03 |
| | VSLNet [50] | - | 45.91 | 19.80 | 41.63 | 24.25 | 11.54 | 31.43 | 25.60 | 10.07 | 30.21 |
| | 2D-TAN [18] | - | 48.06 | 27.10 | 43.72 | 32.74 | 15.25 | 31.50 | 37.12 | 18.99 | 35.04 |
| | 2D-TAN$_{SSL}$ [51] | - | 53.91 | 31.82 | 46.84 | 35.42 | 17.95 | 33.07 | 43.60 | 25.32 | 39.32 |
| | LGI [52] | - | 49.45 | 23.80 | 45.01 | 29.42 | 12.73 | 30.09 | 26.48 | 12.47 | 27.62 |
| | MS-2D-TAN [53] | - | 57.85 | 37.63 | 50.51 | 43.17 | 23.27 | 38.06 | 45.76 | 27.19 | 40.80 |
| | MS-2D-TAN$_{SSL}$ [51] | - | 58.14 | 37.98 | 50.58 | 46.54 | 25.10 | 40.00 | 50.36 | 28.78 | 43.15 |
| | VISA [9] | - | 53.20 | 26.52 | 47.11 | 45.41 | 22.71 | 42.03 | 42.35 | 20.88 | 40.18 |
| | Deco [8] | - | 58.75 | 28.71 | 49.06 | 47.39 | 21.06 | 40.70 | - | - | - |
| | Moment-DETR [54] | - | 49.48 | 28.04 | 44.82 | 39.42 | 18.62 | 36.61 | 46.76 | 24.75 | 41.70 |
| | Moment-DETR$_S$ [1] | - | 57.14 | 33.85 | 49.32 | 44.65 | 23.21 | 39.86 | 47.05 | 24.32 | 41.57 |
| | QD-DETR [55] | 7.12M | 59.24 | 33.43 | 50.22 | 42.30 | 21.09 | 38.55 | 46.04 | 26.33 | 42.89 |
| | QD-DETR$_S$ [1] | 7.12M | 60.66 | 38.60 | 52.53 | 50.23 | 27.69 | 44.14 | 55.25 | 35.25 | 48.10 |
| Weak Supervision | WSSL [31] | - | 15.33 | 5.46 | 18.31 | 3.61 | 1.21 | 8.26 | 2.79 | 0.73 | 7.92 |
| | CNM [10] | 2.52M | 36.37 | 15.25 | 37.88 | 25.04 | 9.12 | 30.79 | 31.37 | 13.24 | 34.38 |
| | CPL [7] | 3.01M | 53.04 | 24.71 | 45.82 | 40.79 | 16.15 | 37.46 | 42.45 | 21.44 | 39.20 |
| | CCR [32] | 9.01M | 50.58 | 24.61 | 45.62 | 39.57 | 16.15 | 37.03 | 41.73 | 21.15 | 38.19 |
| | QMN [6] | 12.51M | 51.65 | 22.64 | 45.85 | 40.67 | 15.72 | 37.91 | 46.91 | 21.58 | **41.07** |
| | PPS [2] | 7.31M | 51.74 | 25.87 | 45.63 | 40.09 | **17.11** | 37.07 | 42.01 | 21.44 | 38.23 |
| | **PC-Net**(Ours) | 3.34M | **54.84** | **26.68** | **47.12** | **41.69** | 16.73 | **38.04** | 46.91 | **23.60** | 41.06 |

Table 2: Comparative performance on ActivityNet-CG datasets.

| | Method | Params | Test-Trivial | | | Novel-Composition | | | Novel-Word | | |
|---|---|---|---|---|---|---|---|---|---|---|---|
| | | | R1@0.5 | R1@0.7 | mIoU | R1@0.5 | R1@0.7 | mIoU | R1@0.5 | R1@0.7 | mIoU |
| Full Supervision | TSP-PRL [22] | - | 34.27 | 18.80 | 37.05 | 14.74 | 1.43 | 12.61 | 18.05 | 3.15 | 14.34 |
| | TMN [49] | - | 16.82 | 7.01 | 17.13 | 8.74 | 4.39 | 10.08 | 9.93 | 5.12 | 11.38 |
| | 2D-TAN [18] | - | 44.50 | 26.03 | 42.12 | 22.80 | 9.95 | 28.49 | 23.86 | 10.37 | 28.88 |
| | LGI [52] | - | 43.56 | 23.29 | 41.37 | 23.21 | 9.02 | 27.86 | 23.10 | 9.03 | 26.95 |
| | VLSNet [50] | - | 39.27 | 23.12 | 42.51 | 20.21 | 9.18 | 29.07 | 21.68 | 9.94 | 29.58 |
| | VISA [9] | - | 47.13 | 29.64 | 44.02 | 31.51 | 16.73 | 35.85 | 30.14 | 15.90 | 35.13 |
| | Deco [8] | - | 43.98 | 24.25 | 43.47 | 27.35 | 11.66 | 31.27 | - | - | - |
| | Moment-DETR [54] | - | 42.73 | 25.31 | 42.19 | 29.29 | 13.71 | 31.63 | 26.84 | 13.34 | 29.95 |
| | Moment-DETR$_S$ [1] | - | 44.19 | 25.81 | 43.49 | 30.60 | 14.40 | 33.13 | 29.59 | 15.10 | 32.43 |
| | QD-DETR [55] | 7.92M | 41.80 | 20.88 | 41.15 | 26.91 | 10.96 | 31.01 | 27.09 | 11.38 | 31.21 |
| | QD-DETR$_S$ [1] | 7.92M | 43.76 | 25.98 | 42.86 | 29.56 | 14.37 | 32.44 | 27.60 | 13.11 | 30.98 |
| Weak Supervision | WSSL [31] | - | 11.03 | 4.14 | 15.07 | 2.89 | 0.76 | 7.65 | 3.09 | 1.13 | 7.10 |
| | CNM [10] | 2.38M | 28.55 | 13.44 | 35.06 | 18.38 | 7.22 | 28.19 | 21.07 | 9.59 | 29.71 |
| | CPL [7] | 4.64M | 27.62 | 11.80 | 32.73 | 19.31 | 7.05 | 26.95 | 22.50 | 9.29 | 28.33 |
| | CCR [32] | 268.96M | 27.67 | 12.90 | 33.56 | 19.59 | 7.66 | 27.51 | 21.66 | 9.18 | 28.42 |
| | QMN [6] | 272.38M | 24.27 | 13.19 | 33.82 | 15.88 | 6.09 | 27.30 | 19.31 | 7.76 | 28.96 |
| | PPS [2] | 8.94M | **30.00** | **15.84** | 32.98 | **20.60** | **9.45** | 26.27 | **22.98** | **11.25** | 27.69 |
| | **PC-Net**(Ours) | 4.97M | 29.62 | 14.35 | **36.45** | 20.16 | 8.05 | **29.51** | 22.88 | 9.85 | **30.76** |

### 4.1 Experimental Setup

**Datasets.** To validate the proposed PC-Net for the WSCMR task, Charades-CG and ActivityNet-CG are applied, where accurate timestamp annotations are only used for evaluating the model rather

than training, and annotations with novel queries are sourced from the literature [9]. Charades-CG (8,312 videos) includes 8,281 training queries and three test subsets: Test-Trivial (3,096 queries with training-style phrases), Novel-Composition (3,442 queries covering verb-noun, noun-noun, verb-adverb, adjective-noun, and preposition-noun combinations [9]), and Novel-Word (703 queries with unseen vocabulary). ActivityNet-CG (20,647 videos) follows a similar split: 36,724 training queries, 15,712 Test-Trivial, 12,028 Novel-Composition, and 3,944 Novel-Word.

**Evaluation Metrics.** Two metrics are introduced. "mIoU" is the mean Intersection over Union (IoU), which reflects the average overlap between predicted timestamps and ground truth, indicating overall performance. "R$n$@$m$" measures recall at top-$n$ predictions under an IoU threshold of $m$.

**Implementation Details.** The implementation details are presented in Appendix A.5.

## 4.2 Comparisons with SOTAs

In addition to recent fully-supervised compositional moment retrieval methods [1, 8, 9], we also reproduce several recent weakly supervised methods from publicly available repositories [2, 6, 7, 10, 32] for comparative evaluation. To fully verify the effectiveness of the proposed method, this paper also provides comparative experiments on testing weakly supervised moment retrieval on two long video datasets (Appendix B.1), model robustness (Appendix B.2) and efficiency (Appendix B.3).

The proposed PC-Net is evaluated on Charades-CG and ActivityNet-CG under weak supervision. As shown in Table 1, PC-Net achieves state-of-the-art performance on Charades-CG, particularly in novel composition and unseen vocabulary scenarios. Under the Novel-Composition split, it obtains R1@0.5/mIoU of 41.69%/38.04%, outperforming prior weakly supervised methods (CPL: 40.79%/37.46%; QMN: 40.67%/37.91%) with only 3.34M parameters—significantly fewer than fully supervised QD-DETR (7.12M) and weakly supervised QMN (12.51M). The performance gain stems from two key innovations: (1) a dual-granularity proposal generator combining global and local multimodal features for robust query understanding and frame-wise temporal association, achieving 54.84% R1@0.5 on Test-Trivial (vs. 53.04% for CPL); and (2) peak-aware feature aggregation with semantic alignment via differentiable weight redistribution and contrastive learning to obtain discriminative proposal features. Notably, PC-Net achieves 89.7% mIoU (47.12%/52.53%) on Test-Trivial with 53.01% fewer parameters than the prior SOTA QD-DETR$_S$.

On the ActivityNet-CG dataset (Table 2), PC-Net continues to outperform existing weakly supervised methods, achieving state-of-the-art mIoU scores across all splits: 36.45% (Test-Trivial), 29.51% (Novel-Composition), and 30.76% (Novel-Word). These results highlight PC-Net's strong semantic alignment capability and query generalization. Although PC-Net's R1@0.5 and R1@0.7 scores are occasionally lower than those of other models on ActivityNet-CG, this is mainly due to the dataset's long average video duration (117.6 seconds) and sparse feature sampling. Nevertheless, its strong mIoU performance confirms the model's capability to fully exploit the query semantics and generate high-quality proposals. Notably, PC-Net remains highly parameter-efficient, requiring only 4.97M parameters—dramatically fewer than CCR (268.86M) and QMN (272.38M). The proposed semantic alignment and quality margin regularizer further enforces the semantic consistency of proposal features and the capability of capturing nuance discrepancy between queries and proposals, making PC-Net a compact yet effective solution for compositional moment retrieval under weak supervision.

Table 3: Detailed ablation experiments are conducted on the Charades-CG dataset.

| Setting | DPG | LPG | $\mathcal{L}_{sem}$ | $\mathcal{L}_{qua}$ | Test-Trivial R1@0.5 | R1@0.7 | mIoU | Novel-Composition R1@0.5 | R1@0.7 | mIoU | Novel-Word R1@0.5 | R1@0.7 | mIoU |
|---|---|---|---|---|---|---|---|---|---|---|---|---|---|
| (a) | | | | | 53.04 | 24.71 | 45.82 | 40.79 | 16.15 | 37.46 | 42.45 | 21.44 | 39.20 |
| (b) | ✓ | | | | 54.39 | 24.87 | 46.89 | 40.97 | 16.40 | 37.77 | 46.06 | 22.32 | 40.98 |
| (c) | | ✓ | | | 51.87 | 23.19 | 45.41 | 40.38 | 16.56 | 37.36 | 42.16 | 22.30 | 38.98 |
| (d) | | | ✓ | | 54.43 | 24.06 | 46.42 | 41.61 | 16.13 | 38.02 | 46.06 | 21.44 | 40.98 |
| (e) | | | | ✓ | 51.52 | 23.55 | 45.66 | 39.80 | 17.00 | 37.57 | 43.60 | 22.16 | 40.07 |
| (f) | ✓ | ✓ | | | 54.07 | 25.65 | 47.08 | 40.74 | 16.64 | 37.72 | 46.20 | 23.45 | 41.04 |
| (g) | ✓ | | ✓ | | 54.04 | 24.94 | 46.59 | 41.48 | 16.91 | 37.49 | 46.91 | 23.60 | **41.17** |
| (h) | ✓ | ✓ | ✓ | | 53.26 | 24.97 | 46.28 | 40.99 | 16.39 | 36.94 | 45.18 | 22.01 | 40.28 |
| (i) | ✓ | ✓ | | ✓ | 54.65 | 25.16 | 46.91 | 40.78 | 16.55 | 37.62 | 45.76 | 21.73 | 39.93 |
| (j) | ✓ | | ✓ | ✓ | 53.88 | 24.52 | 46.61 | 41.04 | 16.82 | 38.01 | 46.19 | 23.60 | 40.41 |
| (k) | | ✓ | ✓ | ✓ | 54.33 | 25.55 | 46.75 | 41.52 | **17.81** | 37.74 | 45.32 | 22.45 | 40.71 |
| Ours | ✓ | ✓ | ✓ | ✓ | **54.84** | **26.68** | **47.12** | **41.69** | 16.73 | **38.04** | **46.91** | **23.60** | 41.06 |

## 4.3 Ablations

Comprehensive ablation studies are conducted on the Charades-CG dataset (Table 3) to assess the impact of each proposed module. Specifically, 'DPG' denotes the dual-granularity proposal generator, and 'PFA' refers to the proposal feature aggregator, which incorporates both cross-modal semantic contrastive loss ($\mathcal{L}_{sem}$) and the learnable peak-aware Gaussian distributor ('LPG'). The contrastive loss in quality margin regularizer is denoted as $\mathcal{L}_{qua}$. Introducing the dual-granularity proposal generator (DPG) (b) consistently improves performance across all metrics compared to the baseline (a), highlighting its role in enhancing temporal perception and proposal boundaries. However, the LPG and quality margin contrastive loss (c, e) slightly degrade performance, possibly due to suboptimal query semantics utilization or re-weighting side effects. Incorporating DPG in settings (j, h) yields performance gains over the full model ("Ours"), underscoring the value of query-driven proposals. Joint DPG and LPG application (f) enhances generalization, particularly on Test-Trivial and Novel-Word splits. Integrating $\mathcal{L}_{sem}$ and $\mathcal{L}_{qua}$ (f&ours) significantly boosts semantic alignment and quality assessment. The full model ("Ours") achieves SOTA results, validating the necessity of each module in addressing the existing limitations in generalizing to compositional queries with diverse temporal and semantic dependencies. More ablation experiments in Appendix B.4.

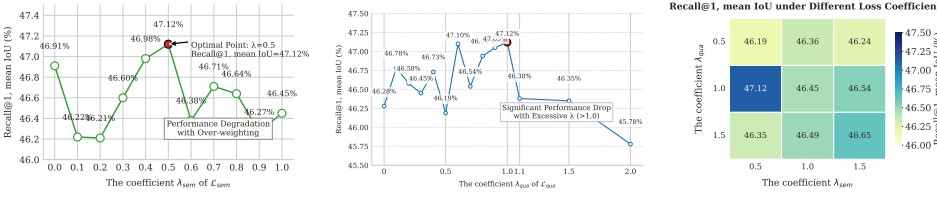

(a) Coefficient Ablation of $\mathcal{L}_{sem}$. (b) Coefficient Ablation of $\mathcal{L}_{qua}$. (c) $\mathcal{L}_{sem}$ and $\mathcal{L}_{qua}$ co-ablation.

Figure 3: Detailed ablation of the alignment loss and quality contrastive loss on Charades-CG.

To deeply analyze the role of the two losses of the proposed semantic alignment loss ($\mathcal{L}_{sem}$) and the quality contrastive loss ($\mathcal{L}_{qua}$), the coefficients of $\mathcal{L}_{sem}$ and $\mathcal{L}_{qua}$ are independently ablated as shown in Figure 3. It can be seen that the model performs best when the coefficient of $\mathcal{L}_{sem}$ is 0.5 and the coefficient of $\mathcal{L}_{qua}$ is 1.0. As the coefficient increases, the model's performance gradually decreases. In addition, to analyze the synergistic effect of the two coefficients, the coefficient combination is also ablated here, thoroughly verifying that the model performs best when $\lambda_{sem}$ is 0.5 and $\lambda_{qua}$ is 1.0.

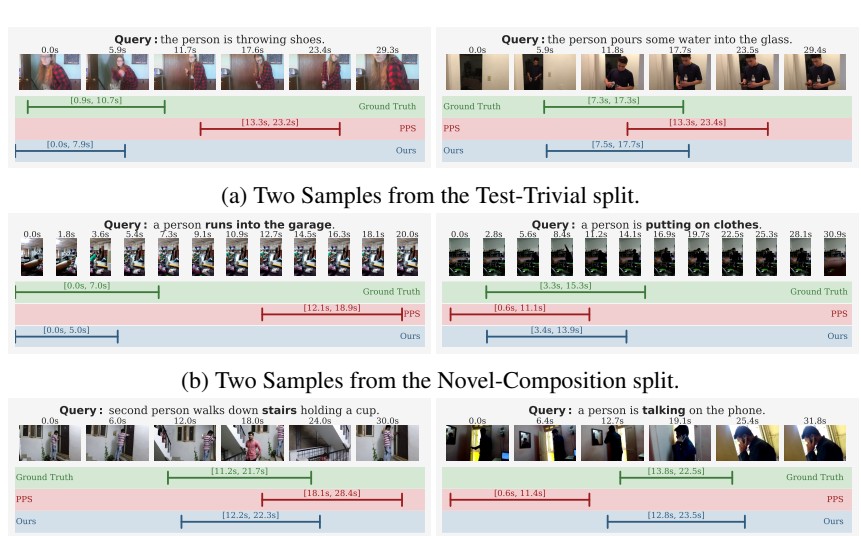

(a) Two Samples from the Test-Trivial split.

(b) Two Samples from the Novel-Composition split.

(c) Two Samples from the Novel-Word split.

Figure 4: Qualitative comparison on Charades-CG dataset.

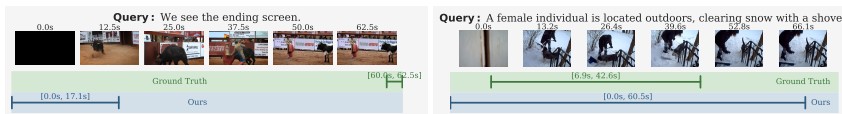

Figure 5: Failure case study on the Test-Trivial split of ActivityNet-CG dataset.

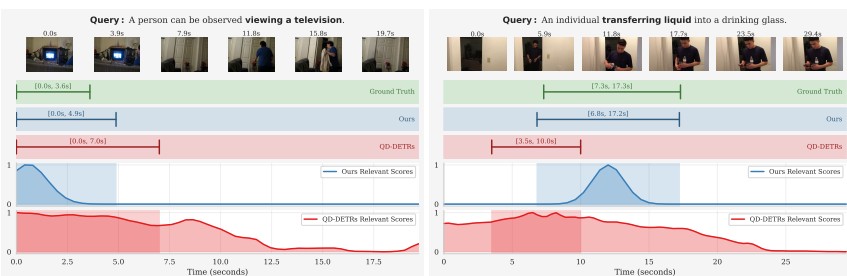

Figure 6: Relevance study on the rewritten Test-Trivial split of Charades-CG dataset.

## 4.4 Visualization

To evaluate the proposed method's retrieval performance, qualitative results on the Charades-CG dataset are presented in Figure 4. In the test-trivial subset (matching the training query style), the method generates high-quality proposals guided by query semantics and selects retrieval results with accurate timestamp alignment via precise boundary modeling. For novel action combinations within compositional queries, Gaussian distribution peak adjustment captures various durations of diverse actions, aggregating them into discriminative features for quality assessment. In the Novel-word scenario (where unseen words are encountered during training), the model leverages deep query associations and explicit semantic alignment to demonstrate robust generalization.

In addition, a failure case study is presented in Figure 5. These examples show that actions with long durations tend to exhibit lower boundary accuracy. This issue arises from the model's inability to accurately capture the association between the start and end points of an action when generating proposals. In future work, we plan to employ more powerful feature extractors, capture query correlations, to adjust and improve proposal boundaries dynamically.

To thoroughly investigate the interpretability of fully supervised versus weakly supervised methods, we conducted visualizations of correlation scores between queries and video frames on the rephrased Charades Test-Trivial dataset, as shown in Figure 6. The results demonstrate that, compared to the fully supervised QD-DETRs trained with precise timestamp annotations, our proposed PC-Net not only comprehensively understands unseen complex query semantics but also more accurately aligns query content with the visual features of video frames. This capability produces more rational and discriminative correlation score curves. Specifically, for the query "A person can be observed viewing a television." PC-Net generates significantly higher correlation scores within the critical action interval [0.0s, 4.9s] than QD-DETRs, indicating a stronger ability to capture the semantics of "viewing a television." Similarly, in the task "An individual transferring liquid into a drinking glass." although QD-DETRs exhibit responses in certain intervals, PC-Net's correlation scores are more concentrated and consistently surpass the baseline in the true action interval [7.3s, 17.3s]. This suggests a more stable alignment between query semantics and relevant visual moments.

## 5 Conclusion

This paper analyzes the shortcomings of existing methods, and proposes a more practical and scalable task, namely WSCMR. By fully mining the dual-granularity query semantics and temporal perception to obtain query-relevant and well-bounded proposals, and improving feature discrimination through the semantic alignment and peak optimization, and the quality margin regularizer is used to establish associations between common visual elements in proposals and queries and to suppress spurious associations, a proposal-centric optimization pipeline is implemented. Extensive experiments demonstrate PC-Net's excellent performance with fewer parameters. Future work will explore improving the query generalization of weakly supervised moment retrieval in long videos.

# 6  Acknowledgements

This work was supported in part by National Natural Science Foundation of China under Grant 62201222 and 62377026, in part by the Fundamental Research Funds for the Central Universities under Grant CCNU25ai041 and CCNU25JC045, and in part by Hubei Provincial Key Laboratory of Artificial Intelligence and Smart Learning under Grant 2025AISL011.

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

# A  More Technical Details

## A.1  Multi-modal Feature Fusion

To effectively model the interaction between video features $\hat{\mathcal{V}}$ and query features $\hat{\mathcal{Q}}$, we propose a two-stage framework that first extracts global query semantics and then establishes both global and local multimodal associations. The pipeline is formalized as follows: the input query features $\hat{\mathcal{Q}} \in \mathbb{R}^{N \times d}$ (where $N$ is the number of query tokens, $d$ is the feature dimension) are augmented with a learnable CLS token $q^{\mathrm{cls}} \in \mathbb{R}^{1 \times d}$ to encode global semantics. The concatenated input $[q^{\mathrm{cls}} \| \hat{\mathcal{Q}}] \in \mathbb{R}^{(N+1) \times d}$ is processed through a self-attention mechanism to generate the global query feature $q^{\mathrm{global}} \in \mathbb{R}^{1 \times d}$ and an encoded query representation $\hat{\mathcal{Q}}^{\mathrm{enc}} \in \mathbb{R}^{N \times d}$. This is mathematically expressed as:

$$q^{\mathrm{global}}, \hat{\mathcal{Q}}^{\mathrm{enc}} = \mathrm{SelfAttn}([q^{\mathrm{cls}} \| \hat{\mathcal{Q}}]), \tag{10}$$

where the CLS token $q^{\mathrm{cls}}$ acts as an aggregator for global query semantics, and the self-attention mechanism computes pairwise attention scores between all tokens in the input sequence. The output $q^{\mathrm{global}}$ captures the global context of the query, while $\hat{\mathcal{Q}}^{\mathrm{enc}}$ retains refined local query features influenced by the global context. The self-attention operation follows the standard formulation:

$$\mathrm{Attention}(Q, K, V) = \mathrm{Softmax}\left(\frac{QK^T}{\sqrt{d}}\right) V, \tag{11}$$

where $Q, K, V$ are linear projections of the input sequence $[q^{\mathrm{cls}} \| \hat{\mathcal{Q}}]$ using learnable weight matrices. This stage ensures that the model learns a compact, global representation of the query while preserving local interactions among query tokens.

In the subsequent cross-attention stage, the input video features $\hat{\mathcal{V}} \in \mathbb{R}^{T \times d}$ (where $T$ is the number of video frames) are also augmented with a learnable CLS token $v^{\mathrm{cls}} \in \mathbb{R}^{1 \times d}$ to capture global association between video and query. Then, the global query feature $q^{\mathrm{global}}$ and encoded query features $\hat{\mathcal{Q}}^{\mathrm{enc}}$ are concatenated as a new query vector $[q^{\mathrm{global}} \| \hat{\mathcal{Q}}^{\mathrm{enc}}] \in \mathbb{R}^{(N+1) \times d}$ to attend over the video features $[v^{\mathrm{cls}} \| \hat{\mathcal{V}}] \in \mathbb{R}^{(T+1) \times d}$ (where $T$ is the number of video frames) through the cross-attention mechanism. This process generates two outputs: $\mathcal{H}^{\mathrm{global}} \in \mathbb{R}^{1 \times d}$, a multimodal feature capturing the global association between the query and video, and $\mathcal{H}^{\mathrm{local}} \in \mathbb{R}^{T \times d}$, frame-wise multimodal features modeling local query-video interactions. The operation is formalized as:

$$\mathcal{H}^{\mathrm{global}}, \mathcal{H}^{\mathrm{local}} = \mathrm{CrossAttn}([v^{\mathrm{cls}} \| \hat{\mathcal{V}}], [q^{\mathrm{global}} \| \hat{\mathcal{Q}}^{\mathrm{enc}}], [q^{\mathrm{global}} \| \hat{\mathcal{Q}}^{\mathrm{enc}}]), \tag{12}$$

where the cross-attention mechanism computes:

$$\mathrm{CrossAttn}(Q, K, V) = \mathrm{Softmax}\left(\frac{QK^T}{\sqrt{d_k}}\right) V, \tag{13}$$

with $Q$ derived from the concatenated video features $[v^{\mathrm{cls}} \| \hat{\mathcal{V}}]$ and $K, V$ as linear projections of $[q^{\mathrm{global}} \| \hat{\mathcal{Q}}^{\mathrm{enc}}]$. The output $\mathcal{H}^{\mathrm{global}}$ aggregates attention-weighted video features to encode holistic relationships for generation proposals with the semantic consistency of the video-query pair, while $\mathcal{H}^{\mathrm{local}}$ preserves frame-level interactions for fine-bounded proposals with fine-grained temporal perception.

## A.2  Details of the Fusion of Global and Local Proposals

Given that single-granularity proposals are subject to corresponding limitations, that is, global proposals $\mathcal{P}^{\mathrm{global}}$ have blurred boundaries due to the inability to perceive frame-level video-query correlation, while local proposals $\mathcal{P}^{\mathrm{local}}_K$ are difficult to avoid interference from irrelevant frame features due to the lack of overall query semantic guidance. Therefore, we propose to use adaptive fusion to alleviate the single-granularity limitation, thereby obtaining a set of proposals with query-related boundaries and temporal logic. First, the Hungarian matching distance [48] is calculated for measuring the correlation between the corresponding proposals of two sets, as follows:

$$\Pi^* = \arg \min_{\Pi \in \mathcal{A}_N} \sum_{i=1}^{N} \left\| \begin{bmatrix} c_i^{\mathrm{global}} \\ w_i^{\mathrm{global}} \end{bmatrix} - \begin{bmatrix} c_{\Pi(i)}^{\mathrm{local}} \\ w_{\Pi(i)}^{\mathrm{local}} \end{bmatrix} \right\|_2, \tag{14}$$

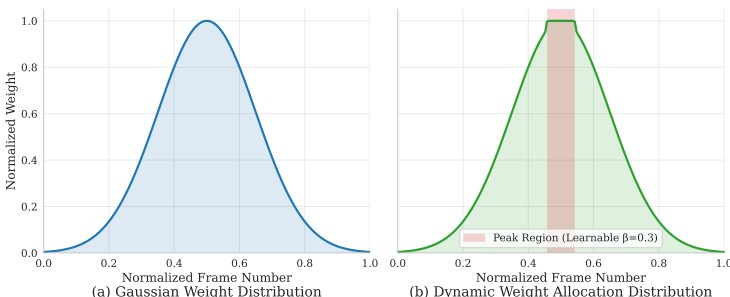

Figure 7: Comparison of the original Gaussian scores and the proposed learnable peak-aware redistribution. $\beta$ is a learnable parameter for controlling peak range to capture diverse durations of various actions in compositional queries.

where $\mathcal{A}_N$ is the permutation and combination space between $\mathcal{P}^{\text{global}}$ and $\mathcal{P}_K^{\text{local}}$, $\Pi^*$ is the optimal matching function. Global-local proposal boundary fusion based on the best match $\Pi^*$.

$$c_i^{\text{fused}} = \sigma(\alpha) \cdot c_i^{\text{global}} + [1 - \sigma(\alpha)] \cdot c_{\Pi^*(i)}^{\text{local}}, \tag{15}$$

$$w_i^{\text{fused}} = \sigma(\alpha) \cdot w_i^{\text{global}} + [1 - \sigma(\alpha)] \cdot w_{\Pi^*(i)}^{\text{local}}, \tag{16}$$

where $\sigma(x) = \frac{1}{1+e^{-x}}$ and $\alpha$ is the learnable coefficient. $\mathcal{P}^{\text{fused}} = \{(c_i^{\text{fused}}, w_i^{\text{fused}})\}_{i=1}^{num_p}$ is the final proposal set injected by global semantic consistency and local temporal dynamics.

### A.3 Original Gaussian Scores Vs. the Proposed Learnable Peak-aware Redistribution

This paper first obtains the corresponding frame-level weights according to the proposal boundary according to the existing work. However, the rigid Gaussian distribution is difficult to fit the duration characteristics of various actions in different compositional queries, resulting in low discriminability of the proposal features aggregated by these weights. Therefore, we propose a learnable peak-aware Gaussian distributor to better cope with the diverse continuous frames of novel action compositions. Gaussian distribution is first used to generate corresponding weights according to the center and duration of each proposal. For the $i$-th proposal in $\mathcal{P}^{\text{fused}} = \{(c_i^{\text{fused}}, w_i^{\text{fused}})\}_{i=1}^{num_p}$ and normalized frame number $x_t$, calculate the Gaussian weights:

$$G_i(t) = \frac{1}{\sigma_i \sqrt{2\pi}} \cdot \exp\left(-\frac{\left(x_t - c_i^{\text{fused}}\right)^2}{2\sigma_i^2}\right), \text{where } x_t = \frac{t-1}{T-1} \ (t = 1, 2, \ldots, T), \tag{17}$$

where $c_i^{\text{fused}}$ is the center point in $\mathcal{P}_i^{\text{fused}}$, $G_i \in \mathbb{R}^T$ is the frame-level weight of the $i$-th proposal and $\sigma_i = \frac{\max\left(w_i^{\text{fused}}, 10^{-2}\right)}{9}$ following existing works [10, 7, 6]. As can be seen from Figure 7, the learnable peak of the model is not a single frame but more potential key frames relevant to compositional queries are included to improve the discriminability of proposal features.

### A.4 Masked Query Reconstruction

To accurately evaluate the quality of proposals, we follow the principle of existing work [7, 35]: the closer the proposal and query are semantically, the higher the proposal quality. Consistent with prior works [10, 6], some words in the original query will be masked and restored through the corresponding proposal features through autoregressive operations. The restored queries by responding proposals will be compared with the original query, and the negative logarithmic loss will be used to quantify the quality of the proposal. Specifically, the words in the original query $S = \{s_i\}_i^N$ are replaced by a specific symbol [MASK] at a ratio of 1/3 to obtain the query $\widehat{S}$ to be reconstructed. The word prediction is made based on before the current word to be reconstructed and the proposal features through autoregression [56]. Finally, the reconstruction loss is used to measure the proposal quality.

$$\mathcal{L}_i^{re} = -\sum_{j=1}^{N-1} \log P\left(s_{j+1} \middle| \hat{\mathcal{V}} \odot W_i, \widehat{S}_{1:j}\right). \tag{18}$$

Among them, $\odot$ is the Hadamard dot product for extracting proposal-specific features based on video features $\hat{\mathcal{V}}$ and $W_i$ is the frame-level weights of $i$-th proposal from the learnble peak-aware Gaussian distributor. $\mathcal{L}_i^{re}$ is the reconstruction loss of the $i$-th proposal, that is, the corresponding quality.

## A.5 Implementation Details

We use GloVe [43] to extract textual features with a hidden dimension of 300. For video features, I3D [44] is used for Charades-CG and C3D [45] for ActivityNet-CG, yielding feature dimensions of 1024 and 500, respectively. The number of proposals is set to 8, and the slot attention module is iterated 4 times. The initial value of the proposal fusion coefficient $\alpha$ is 0.2. The loss coefficient for cross-modal semantic alignment is 0.5, and the margin quality of contrastive loss is 0.146, consistent with $\theta_2$ [7]. We use a batch size of 32 and train for 30 epochs. All experiments are conducted on a single NVIDIA GeForce RTX 4090 GPU. All configuration parameters have been open sourced for easy reproduction, see the supplementary materials.

# B More Experiments

## B.1 Performance Comparison in the Context of Weakly-Supervised Video Moment Retrieval

Table 4: Comparative performance on TACoS [57] and Ego4D [58] datasets. R$n$@$m$ denotes the Recall@$n$ metric under an IoU threshold of $m$. Bold results indicate the best-performing methods, while underlined results represent the second-best methods.

| Method | TACoS | | | | | Ego4D | | | | |
|---|---|---|---|---|---|---|---|---|---|---|
| | Params | R1@0.1 | R1@0.3 | R1@0.5 | mIoU | Params | R1@0.1 | R1@0.3 | R1@0.5 | mIoU |
| CNM [10] | 3.30M | 27.77 | 7.05 | 2.32 | 8.73 | 2.84M | 4.40 | 0.97 | 0.49 | 2.06 |
| CPL [7] | 3.95M | 30.14 | 9.50 | - | 9.52 | 3.72M | 6.12 | 1.23 | - | 2.35 |
| CCR [32] | 17.77M | 31.04 | 7.95 | - | 9.28 | 33.53M | 6.20 | **1.63** | - | 2.41 |
| QMN [6] | 21.18M | **32.59** | 9.07 | - | 9.27 | 36.94M | 6.03 | 1.72 | - | 2.38 |
| PPS [2] | 8.26M | 24.44 | 10.07 | 3.87 | 8.08 | 8.03M | 4.46 | 0.91 | 0.26 | 2.08 |
| Ours | 4.28M | 30.64 | **10.25** | **4.00** | **9.79** | 4.05M | **6.23** | 1.52 | **0.69** | **2.50** |

To further prove the effectiveness of the proposed method, the proposed method is also experimented on two public long video positioning datasets, namely TACoS [57] and Ego4D [58], and the experimental settings are consistent with previous works [6, 59]. The TACoS dataset is derived from the MPII cooking activity dataset [60] and focuses on action recognition in a laboratory kitchen environment. The challenge is to accurately locate the fuzzy temporal boundaries of short-term and subtle actions (such as "cutting vegetable") in long videos, and solve the semantic alignment problem of natural language queries and dynamic visual content. At the same time, it is limited by the high-density annotation of laboratory scenes, and the generalization ability of the model is facing a test; the Ego4D-NLQ dataset contains first-person videos of multiple scenes around the world, requiring the model to handle occlusion, perspective changes, and long-term cross-modal understanding in complex dynamic environments. At the same time, it needs to adapt to the open domain generalization challenges brought by multi-national and multi-cultural differences. We use C3D [45] and SlowFast [61] to extract video features of TACoS and Ego4D respectively.

On TACoS, our method (Ours) achieves R1@0.1 30.64% and mIoU 9.79% with 6.66M parameters, which is better than high-parameter methods (such as QMN's 23.55M parameters only improve R1@0.1 to 32.59%), and significantly outperforms CNM (2.32%) and PPS (3.87%) at high IoU threshold (R1@0.5). On the Ego4D dataset, our method surpasses the same-scale methods (such as CNM's 4.40% and 0.97%) in R1@0.1 (6.23%) and R1@0.3 (1.52%) with 6.42M parameters, while mIoU reaches 2.50%, close to high-parameter methods (such as CCR's 2.41%). Experiments show that the proposed method achieves a balance between parameter efficiency and retrieval precision in long video temporal localization tasks in complex scenes by optimizing the lightweight architecture design, especially in fine-grained action and cross-modal alignment.

Table 5: Comparison of Test-Trivial split rewriting queries based on the Charades-CG and ActivityNet-CG datasets, using Qwen3 for query rewriting to obtain semantically consistent and diverse queries. '†' means the fully supervised method. Bold results indicate the best-performing methods, while underlined results represent the second-best methods.

| Model | Rewritten Query | Charades-CG | | | | ActivityNet-CG | | | |
|---|---|---|---|---|---|---|---|---|---|
| | | R1@0.3 | R1@0.5 | R1@0.7 | mIoU | R1@0.1 | R1@0.3 | R1@0.5 | mIoU |
| QD-DETR$_s^\dagger$ | - | - | 60.66 | 38.60 | 52.53 | - | 43.76 | 25.98 | 42.86 |
| PC-Net(Ours) | | 70.87 | 54.84 | 26.68 | 47.12 | 52.78 | 29.62 | 14.35 | 36.45 |
| QD-DETR$_s^\dagger$ | ✓ | 65.67 | 53.62 | 32.53 | 46.27 | 53.18 | 35.76 | 20.23 | 38.04 |
| CNM | ✓ | 43.86 | 29.46 | 14.02 | 29.49 | 45.86 | 25.23 | _13.43_ | 33.01 |
| CPL | ✓ | 44.64 | 31.88 | 14.50 | 29.85 | 45.77 | 24.00 | 11.60 | 31.26 |
| CCR | ✓ | 59.14 | _45.06_ | _21.38_ | 39.28 | 44.65 | _26.95_ | 10.58 | _33.92_ |
| QMN | ✓ | 55.94 | 40.50 | 18.67 | 37.08 | 45.79 | 24.42 | 13.14 | 33.73 |
| PPS | ✓ | _59.82_ | 43.02 | 20.93 | _39.31_ | _48.36_ | 26.13 | 13.24 | 31.71 |
| PC-Net(Ours) | ✓ | **61.18** | **47.13** | **22.61** | **40.90** | **49.85** | **27.64** | **14.38** | **34.59** |

## B.2 Model Robustness Comparison

To test the robustness of models to diverse query expressions, we have additionally used Qwen3 [62] to perform query rewriting on the Test-trivial set of the most representative Charades-CG and ActivityNet-CG datasets. We will further add these convincing experiments to the original paper to demonstrate the PC-Net's effectiveness. As shown in Table 5, our method maintains good performance under this new expression style, fully demonstrating its generalization ability to different query styles. Even under the more challenging ActivityNet-CG mIoU metric, PC-Net experiences less performance degradation (-1.86%) than the fully supervised method QD-DETR$_s$ (-4.82%) when faced with queries with widely varying expression styles. This is due to the adaptation of the constructed dual-granularity proposal generator to potentially complex temporal logic queries, the good proposal boundaries obtained, and the powerful proposal representation ability of the aligned proposal feature aggregator, which makes the discriminative proposal easy to select. In subsequent work, we will further explore more complex forms of semantic combination.

Here are the query rewriting tips we use to allow Qwen3 [62] to rewrite semantically consistent queries with multiple expression styles.

```python
messages = [
        {
                "role": "system",
                "content": (
                "You are a video content retrieval assistant
                    specialized in query rewriting. "
                "Your task is to generate semantically identical
                    but linguistically diverse versions "
                "of video-related queries while strictly
                    preserving their original meaning."
                )
        },
        {
                "role": "user",
                "content": (
                f"Rewrite the following video moment retrieval
                    query into a semantically equivalent "
                f"but linguistically diverse version.
                    Requirements:\n"
                f"1. Preserve the exact meaning of the original
                    query.\n"
```

```
                        f"2. Alter phrasing (e.g., syntax, vocabulary,
                            active/passive voice).\n"
                        f"3. Ensure compatibility with video content
                            retrieval (e.g., action/object/scene
                            descriptions).\n\n"
                        f"Original Query: \"{original_query}\"\n\n"
                        f"Rewritten Query:"
                        )
                }
        ]
```

## B.3  Model Efficiency Comparison

Table 6: Performance and efficiency comparison on Charades-CG and ActivityNet-CG Test-Trivial sets [9]. FLOPs are measured in giga operations (G), parameters in millions (M), and inference time in milliseconds per sample (ms). Best results are in **bold**.

| Method | Charades-CG | | | | ActivityNet-CG | | | |
|---|---|---|---|---|---|---|---|---|
| | FLOPs ↓ | Params ↓ | Time ↓ | mIoU ↑ | FLOPs ↓ | Params ↓ | Time ↓ | mIoU ↑ |
| CNM [10] | **0.30** | **2.52** | **0.13** | 37.88 | **0.29** | **2.38** | **0.13** | 35.06 |
| CPL [7] | 1.91 | 3.01 | 0.70 | 45.82 | 2.98 | 4.64 | 0.87 | 32.73 |
| CCR [32] | 2.36 | 9.01 | 0.83 | 45.62 | 36.17 | 268.96 | 3.15 | 33.56 |
| QMN [6] | 10.04 | 12.51 | 1.70 | 45.85 | 43.85 | 272.38 | 3.96 | 33.82 |
| PPS [2] | 3.28 | 7.31 | 1.27 | 45.63 | 2.94 | 8.94 | 1.15 | 32.98 |
| PC-Net (Ours) | 1.94 | 3.34 | 1.45 | **47.12** | 3.18 | 4.97 | 2.78 | **36.45** |

As shown in Table 6, our proposed PC-Net demonstrates superior performance across both Charades-CG and ActivityNet-CG benchmarks, achieving state-of-the-art mIoU scores of 47.12% and 36.45%, respectively, while maintaining competitive efficiency. On Charades-CG, PC-Net outperforms the previous best method (QMN [6]) by +1.27% mIoU with 80.7% fewer FLOPs (1.94G vs. 10.04G) and 73.3% fewer parameters (3.34M vs. 12.51M). For ActivityNet-CG, our method surpasses CNM [10] by +1.39% mIoU despite a modest increase in computational cost (3.18G FLOPs vs. 0.29G FLOPs), highlighting its scalability to larger-scale datasets. Notably, PC-Net achieves these gains without excessive parameter growth, retaining model compactness (4.97M parameters) compared to resource-heavy alternatives like CCR [32] (268.96M parameters). The balance between accuracy and efficiency emphasizes the high-quality output of dual-granularity proposal generation fusion and the role of semantic alignment and quality contrast loss in assisting the model to learn efficient parameters for effective cross-modal association.

## B.4  More Hyperparameter Ablation Details

To further investigate the impact of the number of iterations $K$ and the initialization value of the learnable proportional factor $\beta$ on the model performance, we conducted a detailed ablation experiment on Charades-CG (Test-Trivial), as shown in Figure 8. Ablation studies reveal that the optimal number of iterations $K=4$ in the dual-granularity proposal generator balances global coherence and local temporal logic, achieving 47.12% Recall@1 with mean IoU. For the proposal feature aggregator, a peak width coefficient $\beta=0.2$ dynamically adjusts Gaussian distributions to suppress irrelevant frames while preserving key semantic cues, yielding 26.7% R@1 with IoU@0.7. These parameters outperform baselines and demonstrate the necessity of adaptive temporal modeling and feature aggregation in weakly supervised settings, enabling robust generalization to diverse query styles and explicit temporal logic. The proposed framework achieves state-of-the-art performance, highlighting its potential for real-world applications in compositional moment retrieval.

To validate the efficacy of key modules in PC-Net, ablation studies are conducted based on the dynamic coefficients in Figure 9. The dual-granularity proposal generator addresses coarse temporal perception by adaptively fusing global and frame-level boundaries, where the fusion coefficient $\alpha$ exhibits an exponential growth phase (epochs 1-12) followed by a stable phase (epochs 13-30),

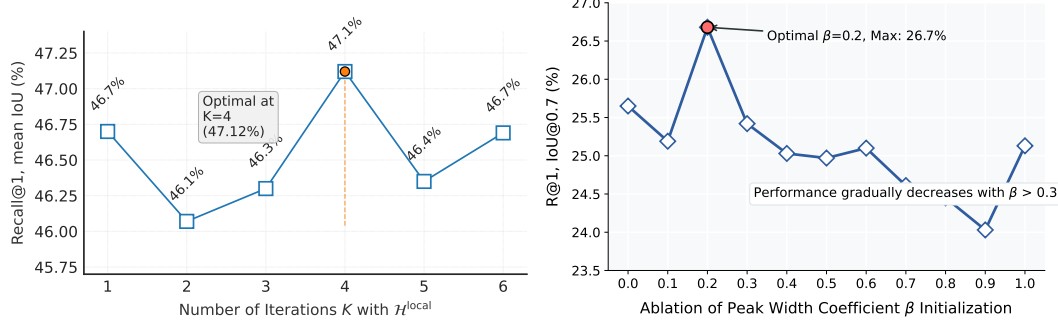

(a) Ablation of the number of iterations $K$.      (b) Ablation of peak width coefficient $\beta$ initialization.

Figure 8: Ablation of the iteration number $K$ for generating local proposals in the dual-granularity proposal generator and the initialization of weight redistribution peak width coefficients $\beta$ in the proposal feature aggregator.

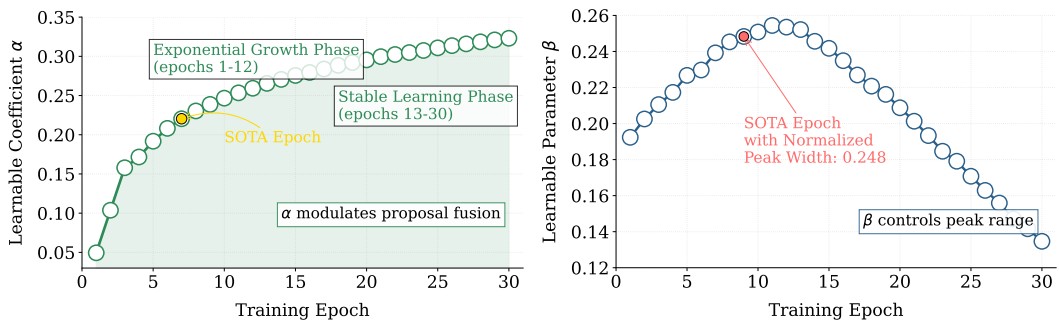

(a) Visualization of dynamic proposal fusion coefficients $\alpha$, based on the ActivityNet-CG dataset [9].

(b) Visualization of learnable peak width coefficients $\beta$, based on the Charades-CG dataset [9].

Figure 9: Visualization of proposal fusion coefficients $\alpha$ in the dual-granularity proposal generator and peak width coefficients $\beta$ in the proposal feature aggregator.

indicating progressive adaptation to query-specific temporal logic. For feature aggregation, the learnable peak width coefficient $\beta$ converges to a normalized width (0.248), demonstrating that the dynamic Gaussian distribution effectively mitigates semantic gaps and captures diverse durations of compositional actions by focusing on query-relevant regions. These results collectively validate that the proposed modules alleviate limitations in boundary generation, feature aggregation, and semantic association modeling under weak supervision.

## C   Limitations

The modular design of PC-Net, with its dynamic boundary modeling and semantic consistency modeling, demonstrates strong cross-task adaptability, enabling direct application to tasks like temporal action localization [63, 64, 65] or event detection [66, 67]. While PC-Net has achieved promising results in weakly supervised compositional moment retrieval (WSCMR), the recall at high IoU thresholds is less impressive in complex dynamic scenes with occlusion or viewpoint changes (e.g., Ego4D). To address these limitations, future work will focus on dense temporal modeling via adaptive frame sampling that prioritizes query-relevant keyframes in long videos, thereby reducing information loss and optimizing proposal boundaries.

