# OpenReview forum: "PC-Net: Weakly Supervised Compositional Moment Retrieval via Proposal-Centric Network"
_NeurIPS.cc/2025/Conference — NeurIPS 2025 poster_

### Official Review · Reviewer_CBCf · 2025-07-01

**Clarity:** 3
**Significance:** 2
**Originality:** 2
**Rating:** 4
**Confidence:** 4

**Summary:**

The paper introduces a new task, Weakly Supervised Compositional Moment Retrieval (WSCMR), in which models are trained only with video–query pairs (no temporal annotations) yet are expected to generalize to queries that contain unseen grammatical structures and vocabulary. To tackle this task the authors propose PC-Net, a “proposal-centric network” that (i) fuses global and frame-level features through a dual-granularity proposal generator, (ii) aggregates frame features with a learnable peak-aware Gaussian distributor and cross-modal contrastive alignment, and (iii) applies a quality-margin regularizer that contrasts high- and low-quality proposals. Experiments on Charades-CG and ActivityNet-CG show that PC-Net attains state-of-the-art mIoU while using fewer parameters than prior work.

**Questions:**

1. Fig. 4 only shows successful cases. Please provide examples where PC-Net fails (e.g., long-range compositional queries) and analyse why.
2. Why is the whole-video reconstruction loss an appropriate decision boundary?
3. What hyper-parameter budget and feature extractors were used when re-running CPL, QMN, PPS, etc.?

**Ethical Concerns:**

["NO or VERY MINOR ethics concerns only"]

**Final Justification:**

Thanks for the authors' detailed response. After careful reading of the rebuttal and comments from other reviewers, I decided to raise my score.

**Limitations:**

Yes

**Quality:**

3

**Strengths And Weaknesses:**

**Strengths:**
1. Solid engineering: each module is motivated by a concrete limitation in prior weak-supervised methods (coarse boundaries, fixed Gaussians, simple negatives).
2. High-level pipeline is clearly illustrated (Fig. 2) and symbols are defined.
3. Raises the bar for weak-supervised moment retrieval by adding compositional generalization—a realistic setting absent from prior work.

**Weaknesses:**
1. While the overall framework is new, the individual components feel like incremental improvements or combinations of existing techniques. The dual-granularity generator's use of global and local features is a common design pattern. The cross-modal semantic alignment (part of PFA) relies on a standard triplet contrastive loss. The "learnable peak-aware Gaussian distributor" is a minor modification to the Gaussian weighting used in prior work. The quality margin regularizer applies another layer of contrastive learning, which has become a standard tool. The novelty appears to lie more in the specific combination of these parts rather than in a truly groundbreaking new mechanism.
2. Some design choices seem heuristic and lack strong theoretical or empirical justification. For instance, the Quality Margin Regularizer contrasts the average reconstruction losses of high- and low-relevance proposal sets. It's unclear why this is superior to methods that contrast individual proposals. Averaging could potentially wash out important learning signals from specific high-quality candidates. The paper does not provide an analysis to justify this specific design choice over simpler alternatives.
3. The ablation study in Table 3 raises some concerns about the stability and contribution of individual modules. When added in isolation, the learnable peak-aware Gaussian (LPG) in setting (c) and the quality margin loss in setting (e) both lead to a decrease in performance on the Novel-Composition split compared to the baseline (a). The authors briefly mention this but do not provide a satisfying explanation. This suggests that the components are not robustly beneficial on their own and that the overall performance gain is the result of a carefully-tuned combination of modules, rather than a set of independently strong contributions.
4. Gains on ActivityNet-CG are modest and sometimes below baselines on recall at stricter IoU—contradicting the “superior” claim.
5. Writing is dense, with long sentences and inconsistent tense; several typos (“quries”) hamper readability.

---

> ### Author Rebuttal · Authors · 2025-07-31
>
> ## Weaknesses
>
> > 1. **While the overall framework is new, the novelty appears to lie more in the specific combination of the proposed parts rather than in a truly groundbreaking new mechanism.**
>
> - First, we clarify that the core contribution of this paper is **the first analysis of potential shortcomings of the existing moment retrieval paradigm in real-world scenarios, proposing the WSCMR, and constructing a simple and effective baseline**.
> - To systematically addresses two core challenges in WSCMR: (1) fine-grained cross-modal alignment under weak supervision, and (2) generalization to compositional queries, the following components are constructed to form PC-Net.
>   - The dual-granularity proposal generator explicitly models complex temporal dependencies (e.g., "after", "second") by jointly decoding frame-level and global features, overcoming existing methods' inability to understand temporal structures in queries.
>   - The proposal feature aggregator employs cross-modal triple contrastive learning to bridge frame-query semantic gaps, while its learnable peak-aware Gaussian distributor dynamically adjusts key area scopes to adapt to variable action durations.
>   - To suppress weak supervision's inherent false correlations, the quality margin regularizer compares high/low relevance proposals, forcing the model to focus on strongly co-occurring visual elements and enhancing robustness to compositional queries.
>
> > 2. **Some design choices appear heuristic and lack strong theoretical or empirical justification (the averaging operation in the quality margin regularizer).**
>
> - First, we have explained the motivation for component design in the introduction and the corresponding method description, and have demonstrated the role of components in promoting the generalization ability of the model through detailed ablation experiments in Table 3, with no lack of evidence.
> - Existing methods overlook that partially relevant proposals can guide the model to capture co-occurring visual elements while suppressing spurious associations. To address this, we introduce a quality margin regularizer ($\mathcal{L} _{\text{qua}}$) as a supplement to the reconstruction loss ($\mathcal{L} _O^{re}$).
> - $\mathcal{L} _{\text{qua}}$ expands the reconstruction loss margin between high- and low-relevance proposal subsets, using the reference proposal as a boundary. This highlights co-occurring elements in high-correlation subsets and suppresses spurious links in low-correlation subsets. As Table 3 (row (h) vs. Ours) shows, optimizing this contrast enhances generalization, confirming $\mathcal{L} _{\text{qua}}$'s effectiveness.
> - The motivation for the construction of the remaining components can be found in the response to weakness 1.
>
> > 3. **The ablation study in Table 3 raises some concerns (robustness) about the stability and contribution of individual modules(setting (c)  and (e)) .**
>
> - Further analysis of the settings follows. Performance slightly declines without the dual-granularity proposal generator due to poor proposal boundaries. This occurs because: (c) Reallocated Gaussian weights cannot adapt to the optimal peak interval within low-quality boundaries, and (e) Such proposals contain irrelevant frames internally while suppressing relevant frames, confusing the model's ability to learn query-relevant moment associations.
>
> - Introducing the proposed dual-granularity proposal generator significantly improves performance (Table 3: (row (j) & Ours) and (row (h) & Ours)), effectively mitigating these limitations of existing methods. Furthermore, robustness analysis in the response to Weakness 4 confirms that PC-Net maintains leading performance even when encountering query expression styles completely dissimilar to those seen during training, demonstrating the inherent robustness of the components.
>
> > 4. **Gains on ActivityNet-CG are modest and sometimes below baselines on recall at stricter IoU—contradicting the “superior” claim.**
>
> Table R1: Comparison of Test-Trivial split rewriting queries based on the Charades-CG and ActivityNet-CG datasets, using Qwen3 for query rewriting to obtain semantically consistent and diverse queries. 'Fully' means the fully supervised method.
>
> |Model|Rewrite queries using Qwen3|Charades-CG||||ActivityNet-CG||||
> |:-:|:-|:-|:-|:-|:-|:-|:-|:-|:-|
> |||**R1@0.3**|**R1@0.5**|**R1@0.7**|**mIoU**|**R1@0.3**|**R1@0.5**|**R1@0.7**| **mIoU**|
> |QD-DETR$_s$(Fully)|No|-|60.66|38.60|52.53|-|43.76|25.98|42.86|
> |PC-Net(Ours)|No|70.87|54.84|26.68|47.12|52.78|29.62|14.35|36.45|
> |QD-DETR$_s$(Fully)|Yes|65.67|53.62|32.53|46.27|53.18|35.76|20.23|38.04|
> |CNM|Yes|43.86|29.46|14.02|29.49|45.86|25.23|13.43|33.01|
> |CPL|Yes|44.64|31.88|14.50|29.85|45.77|24.00|11.60|31.26|
> |CCR|Yes|59.14|45.06|21.38|39.28|44.65|26.95|10.58|33.92|
> |QMN|Yes|55.94|40.50|18.67|37.08|45.79|24.42|13.14|33.73|
> |PPS|Yes|59.82|43.02|20.93|39.31|48.36|26.13|13.24|31.71|
> |PC-Net(Ours)|Yes|**61.18**|**47.13**|**22.61**|**40.90**|**49.85**|**27.64**|**14.38**|**34.59**|
>
> - Thank you for your question. PPS slightly surpasses PC-Net in R1@m through boundary engineering. However, PC-Net has a significant lead in the mIoU metric, which measures the semantic completeness of proposals. Therefore, we use the term "superior". A detailed analysis is as follows.
> - While PPS achieves marginally higher R1@m on ActivityNet-CG, it relies on heavy temporal boundary engineering (redundant proposal enumeration and voting) to inflate recall at the cost of generalizability and efficiency. In contrast, PC-Net excels on the mIoU metric, which directly measures temporal overlap and reflects semantic completeness of retrieved moments. It outperforms PPS by +3.07% to +3.47% mIoU across all ActivityNet-CG splits, demonstrating stronger semantic integrity. Moreover, PC-Net achieves this with half the parameters of PPS while dominating long video benchmarks like TACoS and Ego4D (e.g., +0.43% R1@0.5 on Ego4D), demonstrating its overall superiority.
> - We will refine boundaries through query-guided optimization in future work. This will maximize semantic alignment while preserving efficiency and integrity, ensuring progress on all key metrics.
> - To test model robustness, we used Qwen3 for semantic query rewriting on Charades-CG and ActivityNet-CG datasets. Notably, after LLM-based rewriting of ActivityNet-CG queries, PPS's R1@m dropped sharply, demonstrating its over-reliance on boundary engineering rather than cross-modal semantic associations, which limits adaptability to varied queries.
>
> [1] Siamese learning with joint alignment and regression for weakly-supervised video paragraph grounding.
>
> [2] Query-aware multi-scale proposal network for weakly supervised temporal sentence grounding in videos.
>
> [3] Temporal sentence grounding with relevance feedback in videos.
>
> > 5. **Writing is dense, with long sentences and inconsistent tense; several typos (“quries”) hamper readability.**
>
> - We apologize for the inconvenience caused by this article. The official template is used to write this article. We will further reduce long sentences and fix potential typos to improve readability.
>
> ## Questions
>
> > 1. **Please provide examples where PC-Net fails (e.g., long-range compositional queries) and analyse why.**
>
> - Thank you for your suggestion, we will add the corresponding visualization images in the final version. Here we take the prediction results of ActivityNet-CG after rewriting by Qwen3 as an example, such as the response to Weakness 4.
>
> ```python
> # [vid, duration, GT, query, prediction]
> ["v_Y_dtU10XIsg", 66.06, [6.94, 42.61], "A female individual is located outdoors, clearing snow with a shovel.", [0.0, 60.48]]
> ["v_NMBu3DIn1eg", 202.29, [126.43, 146.66], "The man keeps wiping the chair, making it progressively cleaner.", [49.3, 202.29]]
> ```
>
> - These examples shows that for actions with a long duration, the boundary accuracy is low. This is due to the failure to accurately model the association between the start and end of the action when generating proposals. In the future, we will further try more powerful feature extractors and fully model query correlation, and use it to dynamically adjust and improve the proposal boundary.
>
> > 2. **Why is the whole-video reconstruction loss an appropriate decision boundary?**
>
> - Consistent with previous studies (CPL, CCR, CNM), the full video reconstruction loss, as described in line 210, serves as a suitable decision boundary because it reflects moderate levels of query-correlation: it is weaker than perfectly aligned video moments in terms of semantic relevance, but stronger than completely unrelated video moments [4]. Leveraging this established property, we use the full video loss to split all proposals into high and low query relevance subsets for contrastive learning, capturing query co-occurrence related elements and suppressing potential spurious associations.
>
> [4] Weakly supervised temporal sentence grounding with gaussian-based contrastive proposal learning.
>
> > 3. **What hyper-parameter budget and feature extractors were used when re-running CPL, QMN, PPS, etc.?**
>
> - When reproducing the comparison models such as CPL, QMN, and PPS, we reproduced them completely based on their open source repositories and strictly adopted the hyperparameter settings specified in the corresponding methods to ensure the fairness of the experimental comparison. All weakly supervised methods (including the proposed PC-Net) use the same feature extractor.
> - In addition, the hyperparameter configuration of the proposed PC-Net is also detailed in the implementation details, and the specific hyperparameter settings are provided through the code in the appendix to ensure transparency and consistency.
> - Due to the limited upload capacity of the attachment, all the comparison method reproduction details and the remaining checkpoints will be made public after the paper is accepted.

---

> ### Author Response · Authors · 2025-08-05
>
> Dear reviewer #CBCf:
>
>
>
> Thank you for your valuable time and suggestions regarding visualization and component analysis.
>
>
>
> In our rebuttal, we have clarified the following:
>
> - The core contribution of this paper is the connection between the proposed components and the challenges faced by the WSCMR task
> - The motivation for the quality margin regularizer
> - Analysis of ablation experiments (c)&(e) on some components
> - Detailed experimental analysis on ActivityNet-CG, with additional robustness experiments demonstrating the generalization ability of the proposed PC-Net
> - Failure examples and analysis of PC-Net
> - Explanation of the rationale for using the full video reconstruction loss as the decision boundary
> - Additional description of the experimental setup
>
>
>
> We believe these details have addressed your concerns, and we will incorporate your feedback to further improve the final version.
>
>
>
> We understand that reviewers have limited time to provide feedback; however, we sincerely look forward to your response on rebuttals or any additional insights.
>
>
>
> Thank you again for your comments and valuable time.

---

> ### Author Response · Authors · 2025-08-06
>
> Dear Reviewer #CBCf,
>
> We sincerely appreciate your professional feedback that helped improve the quality of this research.
>
> Given the rigorous nature of academic review, we fully understand that you may need more time to consider your comments.
>
> However, as the discussion window is closing soon, if your concerns have been addressed and there are no further questions, we kindly request you to leave a comment based on our rebuttal.
>
> Best wishes,
> Authors

---

> ### Author Response · Authors · 2025-08-07
>
> Dear reviewer #CBCf,
>
> Thank you for your valuable suggestions.
>
> Due to the rigor of review, we fully understand that it takes time for you to respond to our rebuttal.
>
> However, the discussion window is less than 48 hours away. If you have further questions, please contact us promptly for further clarification.
>
> Thank you again for your time and consideration.
>
> Best wishes.
>
> Authors

---

> ### Author Response · Authors · 2025-08-08
>
> Dear reviewer #CBCf,
>
> Thank you for your valuable suggestions.
>
> Due to the rigor of review, we fully understand that it takes time for you to respond to our rebuttal.
>
> However, the discussion window is less than 24 hours away. If you have further questions, please contact us promptly for further clarification.
>
> Thank you again for your time and consideration.
>
> Best regards.
>
> Authors

---

### Official Review · Reviewer_PDfn · 2025-07-01

**Clarity:** 2
**Significance:** 3
**Originality:** 3
**Rating:** 4
**Confidence:** 3

**Summary:**

The paper proposes the new task of weakly supervised compositional moment retrieval (WSCMR), building on weakly supervised video moment retrieval but applying the models to datasets with with novel syntactic structures or vocabulary. A new model, PC-Net, is proposed to tackle this task, which utilises a number of modules. It uses a dual-granularity proposal generator to produce proposals using a combination of global and local knowledge. It has a proposal feature aggregator with semantic alignment which includes the ability to adjust the Gaussian peak to account for diverse durations. Cross-modal semantic alignment is used to relate the video and text features. The proposals are assessed using query mask reconstruction, and it includes a quality margin regularizer to identify subtle semantic differences between query-relevant proposals. Performance is measured on the Charades-CG and ActivityNet-CG datasets. It achieves broadly improved performance over existing weakly supervised methods, albeit PPS is stronger on many metrics on the ActivityNet-CG benchmark.

**Questions:**

- (line 215) Is the complement just the part of the video outside of the proposal? So is this why there are two parts to the complement (n1 and n2), one for either side of the proposal?
- Overall, while I think this is a promising paper and a promising task, the main issues which I have are with readability and the difficulty in understanding parts of the paper, along with some concerns over performance on ActivityNet-CG on R1@m compared to PPS.

**Ethical Concerns:**

["NO or VERY MINOR ethics concerns only"]

**Final Justification:**

The rebuttal addressed my main concerns. I feel the paper could do with some improved writing to improve the ease of understanding, and the extra information from the rebuttal should be included in a final version. The results, while still weaker on some metrics for ActivityNet-CG, are still strong enough I think, taking into consideration their reasoning for the difference in performance between PC-Net and PPS on the different metrics. As a result, I raise my rating to borderline accept.

**Limitations:**

Yes

**Quality:**

2

**Strengths And Weaknesses:**

**Strengths**
- It is beneficial to attempt the Compositional Video Moment Retrieval task in a weakly supervised manner to avoid the need for extensive annotations. Good from a real-world practicality perspective
- The components generally make sense in accordance with the aim of improving the model's ability to handle diverse moment durations.
- Performance on Charades-CG is strong across all metrics, while performance on ActivityNet-CG is also quite strong.
- The ablations are thorough and display the importance of each component.
- The model shows good efficiency compared with previous methods

**Weaknesses**
- The method contains many parts and it is a bit difficult to understand parts of the paper
- For example, equations 5&6 for the redistribution of weights of the Gaussian distribution. I found it difficult to understand why it is formulated this way, or what the motivation for these equations having this form is. Line 194-196 does not make sense to me grammatically or semantically, "η_i(t) through the distance discrimination mechanism, the peak area and the edge area are divided to achieve dynamic weight allocation". It doesn't help that σ_i and x_t are also not described and only defined in the appendix. Even in the appendix, the purpose of σ_i is not really explained.
- In line 154, c and w are not defined. They are defined much later in the paper but at this point it is potentially confusing.
- I also found the lack of definition of the complement (line 215) and the origin of the two negative proposal reconstruction losses (line 217) to be confusing.
- PPS consistently outperforms PC-Net on R1@m metrics on ActivityNet-CG, which perhaps indicates that PC-Net is not as good at achieving tight moment localisations on this dataset.
- (minor) Line 223 it should say Figure 2(c) rather than Figure 2(e) I assume.

---

> ### Author Rebuttal · Authors · 2025-07-31
>
> ## Weaknesses
>
> > 1. **The method contains many parts and it is a bit difficult to understand parts of the paper.**
>
> - Thank you very much for your careful review and valuable suggestions. We will make revisions based on your suggestions to improve readability and clarify the contributions.
>
> > 2. **Questions related to equations (5)-(6).**
>
> - Due to limited space, the wording in the original text may have been confusing, and the following points are clarified in detail.
> - **Motivation of equations (5) and (6):** Existing methods only use a fixed Gaussian distribution to generate proposal weights, which ignores the diverse durations of potential actions in queries, resulting in insufficient discriminative power of aggregated proposal features from video features. The core function of formulas (5) and (6) is to dynamically redistribute the standard Gaussian weights, to assign higher weights near the center of the proposal (peak area) where the action is most likely to occur, forming a "platform" instead of a sharp single-point peak in the standard Gaussian. Direct hard assignment to achieve peak platform effect is non-differentiable and difficult to optimize through training. Therefore, we introduce a variable $η_i(t)$ with the learnable factor $\beta$. The original formula and explanation are as follows.
>      $$M_i(t) = \frac{1}{1 + e^{-1000 \cdot \eta_i(t)}}, \text{where}\ \eta_i(t) = \beta \sigma_i - |x_t - c_i^{\text{fused}}| \tag{5}$$
>      $$W_i(t) = G_i(t) \cdot (1 - M_i(t)) + M_i(t)  \tag{6}$$
> - **Clarification of $\sigma_i$ and $x_t$ :** $\sigma_i$ and $x_t$ are standard variables in the standard Gaussian distribution, with definitions consistent with prior methods like CPL [1]. $\sigma_i  = \frac{\max\left(w_i^{\text{fused}}, 10^{-2}\right)}{9}$ originates from the fused proposal width $w_i^{\textrm{fused}}$ and governs the shape and width of the Gaussian curve: longer proposals (larger $w_i^{\textrm{fused}}$) result in larger $\sigma_i$ and flatter curves, while shorter proposals yield smaller $\sigma_i$ and sharper curves. $x_t  = \frac{t - 1}{T - 1}\ (t = 1, 2, \ldots, T)$ acts as an auxiliary variable representing positions along the frame sequence axis, used to compute the Gaussian weight at a specific time step $t$.
>
> - **Clarification of $η_i(t)$ and "distance discrimination mechanism":**
>     - $η_i(t) = βσ_i - |t - c^{\textrm{fused}}_i|$ implements a temporal distance discrimination mechanism
>       - $c^{\textrm{fused}}_i,w_i^{\textrm{fused}}$ is the temporal center and width of the $i$-th proposal
>       - $|t - c^{\textrm{fused}}_i|$ computes absolute temporal distance between temporal point $t$ and $c^{\textrm{fused}}_i$
>       - $βσ_i$ sets an adaptive peak range with the learnable $\beta$ where: $\sigma_i = \frac{\max(w_i^{\textrm{fused}}, 10^{-2})}{9}$ ensures numerical stability and scales the fused weight $w_i^{\textrm{fused}}$
>   - **Mask dynamics ($M_i(t)$)**:
>     $M_i(t) = \frac{1}{1+e^{-1000\cdotη_i(t)}}$ creates a differentiable mask:
>     - When $t$ approaches $c^{\textrm{fused}}_i$ (small $|t - c_i^{\textrm{fused}}|$):
>       - $η_i(t) > 0$ → $M_i(t) \approx 1$ tends to assign the maximum weight 1
>     - When $t$ nears video edges (large $|t - c_i^{\textrm{fused}}|$):
>       - $η_i(t) < 0$ → $M_i(t) \approx 0$ tends to preserve the original standard Gaussian weight
>     - The scaling factor 1000 ensures sharp transitions.
>   - **Dynamic weight redistribution ($W_i(t)$)**:
>     $W_i(t) = G_i(t) \cdot (1 - M_i(t)) + M_i(t)$ combines:
>     - **Standard Gaussian weights** $G_i(t)$ calculation following previous methods (CPL,CCR)
>     - **Peak range controlled by $\beta$**:
>       - Near center ($M_i(t) ≈ 1$): $W_i(t) ≈ 1$ (forms "peak area")
>       - At edge ($M_i(t) ≈ 0$): $W_i(t) ≈ G_i(t)$ (preserves standard Gaussian weights)
>     - As visualized in Appendix Figure 5, this mechanism transforms standard Gaussian weights into redistributed weights with enhanced central emphasis, enabling capture of diverse action durations with the learnable $\beta$.
>
>   [1] Weakly supervised temporal sentence grounding with gaussian-based contrastive proposal learning (CVPR'22)
>
> > 3.  **In line 154, c and w are defined later in the paper and it is potentially confusing.**
>
> - Since the proposal boundary is not used immediately after generation and involves subsequent feature aggregation, the definitions of c and w are given in line 183. We will add symbolic explanations to alleviate confusion.
>
> > 4. **Lack of definition of the complement (line 215) and the origin of the two negative proposal reconstruction losses (line 217) to be confusing.**
>
> - Each proposal is defined by its center point ($c_i$) and duration ($w_i$) (line 214). As shown in Figure 2(c), the **complement** of the best proposal $\mathcal{P}^{\text{fused}}_O$ consists of two segments:
>   - $\mathcal{P}^{\text{fused} -l}_O$ (left negative proposal)
>   - $\mathcal{P}^{\text{fused} -r}_O$ (right negative proposal)
>
> - The **negative proposal reconstruction losses** ($\mathcal{L} _{n_1}^{re},\mathcal{L} _{n_2}^{re}$; line 217) are computed for these left/right complements:
>   - Features are aggregated for $\mathcal{P}^{\text{fused} -l}_O$ and $\mathcal{P}^{\text{fused} -r}_O$ using their center points and widths (durations).
>   - Reconstruction loss follows prior work (e.g., CNM, CPL; line 213) to get $\mathcal{L} _{n_1}^{re},\mathcal{L} _{n_2}^{re}$.
> - We will explain its specific meaning more clearly in the final version (i.e., the set of negative proposals on the left and right ends), and strengthen the correspondence with Figure 2(c).
>
> > 5. **Performance comparison of the R1@m metric on ActivityNet-CG.**
>
> Table R1: Comparison of Test-Trivial split rewriting queries based on the Charades-CG and ActivityNet-CG datasets, using Qwen3 for query rewriting to obtain semantically consistent and diverse queries. 'Fully' means the fully supervised method.
>
> |Model|Rewrite queries using Qwen3|Charades-CG||||ActivityNet-CG||||
> |:-:|:-|:-|:-|:-|:-|:-|:-|:-|:-|
> |||**R1@0.3**|**R1@0.5**|**R1@0.7**|**mIoU**|**R1@0.3**|**R1@0.5**|**R1@0.7**| **mIoU**|
> |QD-DETR$_s$(Fully)|No|-|60.66|38.60|52.53|-|43.76|25.98|42.86|
> |PC-Net(Ours)|No|70.87|54.84|26.68|47.12|52.78|29.62|14.35|36.45|
> |QD-DETR$_s$(Fully)|Yes|65.67|53.62|32.53|46.27|53.18|35.76|20.23|38.04|
> |CNM|Yes|43.86|29.46|14.02|29.49|45.86|25.23|13.43|33.01|
> |CPL|Yes|44.64|31.88|14.50|29.85|45.77|24.00|11.60|31.26|
> |CCR|Yes|59.14|45.06|21.38|39.28|44.65|26.95|10.58|33.92|
> |QMN|Yes|55.94|40.50|18.67|37.08|45.79|24.42|13.14|33.73|
> |PPS|Yes|59.82|43.02|20.93|39.31|48.36|26.13|13.24|31.71|
> |PC-Net(Ours)|Yes|**61.18**|**47.13**|**22.61**|**40.90**|**49.85**|**27.64**|**14.38**|**34.59**|
>
> - PPS employs a computationally intensive strategy based on complex, redundant boundary enumeration (predicting i+1 potential boundaries for the i-th proposal, followed by voting) to inflate R1@m. This brute-force approach captures potential target regions at considerable computational expense, and compromises the quality of individual proposals, thereby undermining semantic completeness (as evidenced by lower mIoU).
>
> - In contrast, PC-Net prioritizes semantic association between query and video frames while PPS focuses on boundary engineering. Here, PC-Net dominates: it achieves +3.07% to +3.47% higher mIoU than PPS across all ActivityNet-CG splits (Table 2), proving superior semantic completeness. Critically, PPS's redundant boundary engineering and voting backfire on other long-video benchmarks where semantic coherence matters most. Furthermore, as shown in Appendix B.1, on the TACoS and Ego4D datasets, PC-Net has only half the parameters of PPS, but its performance is slightly better (e.g., R1@0.5 is +0.43% on Ego4D) in Table 4. This efficiency and scalability demonstrate PC-Net's purer cross-modal modeling, while PPS tends to boundary engineering.
>
> - To test the robustness of models to diverse query expressions, we have additionally used the Qwen3 to perform query rewriting on the Test-trivial set of Charades-CG and ActivityNet-CG datasets. As shown in Table R1,it should be emphasized that after the queries in ActivityNet-CG were semantically rewritten by LLM, the R1@m indicator of PPS also dropped sharply, which fully proves that PPS focuses on boundary engineering rather than the more important cross-modal semantic associations, making it difficult to adapt to diverse queries.
>
> - Future work will refine boundaries only when robust query-proposal semantic alignment is first maximized, ensuring efficiency and semantic integrity without resorting to exhaustive enumeration.
>
>
> > 6. **(minor) Line 223 it should say Figure 2(c) rather than Figure 2(e) I assume.**
>
> - Thanks for pointing this out, we will make further corrections.
>
> ## Questions
>
> > 1. **(line 215) Is the complement just the part of the video outside of the proposal?**
>
> - That's correct. The definition of complement is the area outside the target area (proposal) relative to the full set (video), located on both sides of the proposal.
>
> > 2. **While I think this is a promising paper and a promising task, the main issues are with readability and the difficulty in understanding parts of the paper, along with some concerns over performance on ActivityNet-CG.**
>
> - Thanks for your affirmation about research prospects and proposed methods. Due to limited space, some concepts are not introduced in detail, which makes it difficult to understand. We will add relevant explanations of formulas (5) and (6) and the clear definition of the complement of line 215.
> - For more details on the PPS experimental analysis, please see the response to Weakness 5. In short, PPS leverages boundary engineering, achieving a slight lead in R1@m over ActivityNet-CG. However, this introduces significant computational overhead and weakens cross-modal semantic learning. When faced with rewritten queries or other scenarios, PPS performs poorly, while PC-Net maintains strong generalization capabilities.

---

> > ### Comment · Reviewer_PDfn · 2025-08-04
> >
> > Thank you for the detailed response. The explanations of equations 5&6 are much clearer now. Please include this level of detail in the final version of the paper, be it in the main paper or the Appendix.
> >
> > Thank you also for elaborating on the complement and reconstruction losses.  It would help if the subscripts $n_1$ and $n_2$  are explicitly associated with negative 1 and negative 2. Likewise if the subscript $r$ is explicitly associated with the reference proposal.
> >
> > The explanation of the ActivityNet-CG results vs PPS seems reasonable. The additional results using rewritten queries are interesting. I would be interested to see the prompts used to rewrite the queries, along with some examples of cases where model predictions change with rewritten prompts. If that would be feasible it would be a nice addition to the Appendix/supplementary. I feel that I am sufficiently happy that performance is shown to be effective compared to PPS.
> >
> > A small point that I missed before - bolding of top results from other methods is missing in Table 4 of the Appendix.
> >
> > I currently plan to increase my rating.

---

> > > ### Author Response · Authors · 2025-08-04
> > >
> > > - Dear reviewer #PDfn, thank you for your further feedback. We will incorporate your suggested details for the descriptions of Equations 5 & 6, as well as the proposed improvements to the notation of complement and reconstruction losses, into the final version.
> > > - Regarding the details of the robustness experiment, the Qwen3 prompts for rewriting queries and examples of rewritten queries are provided below:
> > >
> > > ```python
> > > messages=[
> > >             {
> > >                 "role": "system",
> > >                 "content": "You are a video content retrieval assistant specialized in query rewriting. "
> > >                            "Your task is to generate semantically identical but linguistically diverse versions "
> > >                            "of video-related queries while strictly preserving their original meaning."
> > >             },
> > >             {
> > >                 "role": "user",
> > >                 "content": f"Rewrite the following video moment retrieval query into a semantically equivalent "
> > >                            f"but linguistically diverse version. Requirements:\n"
> > >                            f"1. Preserve the exact meaning of the original query.\n"
> > >                            f"2. Alter phrasing (e.g., syntax, vocabulary, active/passive voice).\n"
> > >                            f"3. Ensure compatibility with video content retrieval (e.g., action/object/scene descriptions).\n\n"
> > >                            f"Original Query: \"{original_query}\"\n\n"
> > >                            f"Rewritten Query:"
> > >             }
> > >         ],
> > > ```
> > >
> > > | Original                        | rewritten                                                |
> > > | ------------------------------- | -------------------------------------------------------- |
> > > | person pouring it into a glass. | An individual transferring liquid into a drinking glass. |
> > > | person put the clothes.         | Someone places the garments.                             |
> > >
> > > - Below is a comparison of the retrieval results before and after rewriting the query. We will add visualization results to the appendix to demonstrate the effectiveness of the method.
> > >
> > > | Method | Rewritten | Query                      | Ground Truth | Prediction    |
> > > | ------ | --------- | -------------------------- | ------------ | ------------- |
> > > | PPS    | No        | a person is sneezing.      | [0.1, 7.5]   | [0.68, 11.7]  |
> > > | PC-Net | No        | a person is sneezing.      | [0.1, 7.5]   | [0.0, 8.10]   |
> > > | PPS    | Yes       | A person sneezes.          | [0.1, 7.5]   | [0.69, 12.69] |
> > > | PC-Net | Yes       | A person sneezes.          | [0.1, 7.5]   | [0.0, 8.11]   |
> > > | PPS    | No        | a man holds a pillow.      | [0.0, 13.4]  | [0.73, 10.88] |
> > > | PC-Net | No        | a man holds a pillow.      | [0.0, 13.4]  | [0.0, 10.94]  |
> > > | PPS    | Yes       | A pillow is held by a man. | [0.0, 13.4]  | [5.04, 15.3]  |
> > > | PC-Net | Yes       | A pillow is held by a man. | [0.0, 13.4]  | [0.0, 10.41]  |
> > >
> > > - As can be seen, the proposed PC-Net is more robust to novel query styles than PPS, further demonstrating the effectiveness of the proposed components.
> > > - Thank you for pointing out the omission in Appendix Table 4; we will address this in the final version.
> > > - Thank you again for your time and appreciation of this work. We will incorporate all your valuable feedback into the final version to further improve the quality of the paper!

---

> > > ### Author Response · Authors · 2025-08-06
> > >
> > > Dear reviewer #PDfn,
> > >
> > > Thank you for your time and valuable feedback.
> > >
> > > Based on your suggestions, we have provided the setup for the robustness experiment, along with a comparative analysis of the prediction results. As the discussion deadline approaches, please let us know if you have any further concerns or require additional clarification.
> > >
> > > Thank you again for your time and constructive feedback.
> > >
> > > Best wishes,
> > > Authors

---

> > > > ### Comment · Reviewer_PDfn · 2025-08-07
> > > >
> > > > Thanks for the extra details on the prompt for the rewritten queries and some examples. It is interesting to see the differing predictions with the rewritten queries. I don't have any further questions. I will raise my rating.

---

> > > > > ### Author Response · Authors · 2025-08-07
> > > > >
> > > > > Dear reviewer #PDfn:
> > > > >
> > > > > Thank you for recognizing our work and for the rating increase.
> > > > >
> > > > > We're pleased to hear that our clarifications addressed your concerns and curiosity, and we will incorporate these suggestions into the final version to improve its quality.
> > > > >
> > > > > Thank you again for taking the time to review our rebuttal and provide your final comments.
> > > > >
> > > > > Best wishes,
> > > > >
> > > > > Authors

---

> ### Author Response · Authors · 2025-08-04
>
> Dear reviewer #PDfn,
>
> Thank you for your careful review and constructive feedback. We have provided in-depth explanations on equations (5)&(6) and some notational explanations, as well as additional experiments and performance analysis on the ActivityNet-CG dataset, which you are concerned about. We hope that we have addressed most of your concerns, and we plan to incorporate these improvements into the final version. Due to the limited time for discussion, please do not hesitate to let us know if you have any final questions or comments.
>
> We fully understand that reviewers have limited time to provide feedback; however, we sincerely look forward to your feedback on the rebuttal or any additional insights you may offer.
>
> Thank you again for your appreciation of the promise of the task and the model's performance. Thank you for your time and consideration.

---

### Official Review · Reviewer_H739 · 2025-07-02

**Clarity:** 3
**Significance:** 3
**Originality:** 3
**Rating:** 5
**Confidence:** 3

**Summary:**

The paper introduces a proposal-centric network (PC-Nets) for locating video moments in long videos according to natural language queries. For better retrieval of video moments according to queries with novel structure and vocabulary, the PC-Net features a dual-grained proposal generator, a learnable peak-aware Gaussian distributor, and a quality margin regularizer. The proposed model has been extensively evaluated on multiple benchmark datasets across various settings. It is shown that the PC-Net achieves competitive performance to existing solutions with fewer parameters.

**Questions:**

I am generally happy with this submission. It will be helpful if the authors can clarify why their proposed model is targeting WSCMR rather than general VMR as suggested in the weaknesses section.

**Ethical Concerns:**

["NO or VERY MINOR ethics concerns only"]

**Final Justification:**

My main concerns about the unique challenges of WSCMR and how they are addressed by the proposed designs are resolved by the authors. It would be helpful to the clarity if the authors could incorporate their clarification about which components are proposed in this work and which are borrowed in their final version.

**Limitations:**

Yes

**Quality:**

4

**Strengths And Weaknesses:**

Strengths:
+ The paper is well written and easy to follow
+ The weakly-supervised compositional moment retrieval studied in this paper is challenging but practical, which requires generalisation to unseen scenarios while avoiding labour-intensive annotations of moment boundary for training
+ The proposed method is technically solid and well justified. Various components and designs have been made to improve video moment retrieval from different perspectives, from proposal generation to cross-modal feature aggregation and proposal quality assessment.
+ The experiments are comprehensive with detailed ablation studies.

Weaknesses:
+ About motivation, the WSCMR task is explicitly targeted in this paper but it seems like most proposed designs are also valid for general video moment retrieval (VMR). I failed to find strong connections between the designs of PC-Net and the unique challenges of WSCMR. It might be helpful if the authors can spell out and briefly summarise how the challenges unique to WSCMR are addressed by the proposed model. Otherwise, it should be ok to expand the scope to general VMR instead of a specific scenario.
+ About clarity, given that PC-Net is composed of both common components widely adopted in the field and newly proposed ones, I found it hard to identify the latter, given all components are elaborated to a certain extent.

---

> ### Author Rebuttal · Authors · 2025-07-31
>
> ## Weaknesses
>
> > 1. **I failed to find strong connections between the designs of PC-Net and the unique challenges of WSCMR. It might be helpful if the authors can spell out and briefly summarise how the challenges unique to WSCMR are addressed by the proposed model. Otherwise, it should be ok to expand the scope to general VMR instead of a specific scenario.**
>
> - Thank you for your suggestion. First, the relationship between the proposed PC-Net and the challenges faced by WSCMR is analyzed. The challenges of the proposed WSCMR are: **(1)** only relying on weak paired video-query supervision to model fine-grained cross-modal semantic associations; **(2)** the query styles in the training data are limited, and the model needs to be generalized to compositional queries containing novel syntactic structures, unseen vocabulary, and complex temporal semantic dependencies. PC-Net is designed to directly respond to these challenges:
>   - **Coping with diverse syntax and temporal dependencies (Challenge 1&2):**  the dual-granularity proposal generator generates proposal boundaries with fine-grained temporal awareness by jointly decoding frame-level and global multimodal features. This enables it to effectively understand the explicit temporal logic in compositional queries (such as "the second one", "after..."), overcoming the defects of existing methods that rely only on global associations and have difficulty in handling such complex structures.
>   - **Mitigating semantic gap and capturing actions with flexible durations (Challenge 1&2):**  the proposal feature aggregator with semantic alignment employs triples of query-related/unrelated frames to align cross-modal features in a unified semantic space, thereby narrowing the semantic gap. Second, it dynamically adjusts the Gaussian peak area to flexibly adapt to the variable duration of actions described by compositional queries. This ensures that the aggregated proposal features are discriminative and aligned with the query features.
>   - **Learning visual-semantic associations and suppressing spurious correlation (Challenge 1):** the quality margin regularizer helps the model capture co-occurring relevant visual elements in proposals while suppressing potential spurious associations by partitioning proposals into high/low correlation subsets and performing contrastive learning. This indirectly optimizes proposal features modeling and cross-modal fine-grained associations, alleviating the difficulty of cross-modal semantic associations in the absence of precise timestamp associations.
> - Thus, each core component of PC-Net closely corresponds to the challenges of WSCMR. Experimental results verify the effectiveness and efficiency of this design. We will outline this correspondence more clearly in the manuscript to clarify this potential confusion.
> - Besides, this paper mainly emphasizes that the WSCMR task avoids laborious timestamp annotation and has good generalization ability for diverse queries, so it has a broader research prospect than general VMR, rather than referring to "specific scenarios". Lastly, the proposed PC-Net does not contain modules optimized for using precise timestamps and is therefore not suitable for the general VMR.
>
> > 2. **About clarity, given that PC-Net is composed of both common components widely adopted in the field and newly proposed ones, I found it hard to identify the latter, given all components are elaborated to a certain extent.**
>
> - Thank you for your question about the clarity of the model components, which helps us to explain our work more accurately. First, PC-Net integrates common components widely used in the field for proposal quality assessment. Specifically, to compare fairly with existing weakly supervised methods, we use a standard Transformer decoder to reconstruct the masked query and use the corresponding reconstruction loss to assess the proposal quality.
> - The primary innovation of this study resides in analyzing the shortcomings of the existing moment retrieval paradigm and proposing the weakly supervised compositional moment retrieval (WSCMR) task and the construction of three novel modules designed to address the distinct challenges of WSCMR as described in the response to Weakness 1.
> - These modules distinguish our approach from prior work:
>   - The dual-granularity proposal generator models frame-level associations and captures temporal information to refine proposal boundaries.
>   - The proposal feature aggregator that enhances proposal representations through bridging the cross-modal semantic gap and modeling diverse action durations.
>   - The quality margin regularizer that emphasizes relevant visual elements co-occurring in proposals while suppressing spurious associations.
>
> ## Questions
>
> > 1. **I am generally happy with this submission. It will be helpful if the authors can clarify why their proposed model is targeting WSCMR rather than general VMR as suggested in the weaknesses section.**
>
> - Thank you for recognizing the efforts made in this article. We clarify that there are two main reasons for focusing on WSCMR:
>   - General VMR relies on precise timestamp supervision during training, which limits its applicability in such real-world scenarios where annotations are scarce.
>   - Existing weakly supervised VMR methods are difficult to generalize to unseen queries with complex syntactic and temporal logics, which hinders their practical effectiveness.
> - WSCMR addresses these deficiencies by eliminating the need for timestamp supervision and explicitly enhancing the compositional generalization ability to new queries, making it a more promising research direction than VMR. Therefore, the design of the proposed PC-Net is tailored to the core challenges of WSCMR  as described in the response to Weakness 1. We will further highlight the close connection between the proposed components and the WSCMR task in the final version. Thank you again for your high recognition of this article.

---

> > ### Comment · Reviewer_H739 · 2025-08-05
> >
> > Thanks the authors for the detailed explanation about the unique challenges of WSCMR and how they are connected with the proposed designs. I believe it would be helpful to underscore these connections in the next version.
> >
> > Given my main concern is properly addressed by the clarification from the authors, I would still recommend an acceptance for this submission.

---

> > > ### Author Response · Authors · 2025-08-05
> > >
> > > Dear Reviewer #H739:
> > >
> > > Thank you for taking the time to review our rebuttal. We are pleased to hear that our clarifications address your concerns, and we will incorporate these suggestions into the final version to improve its quality.
> > >
> > > Thank you again for your valuable feedback and time.
> > >
> > > Best wishes.

---

> ### Author Response · Authors · 2025-08-05
>
> Dear reviewer #H739,
>
> Thank you for your careful review and high recognition of this submission.
>
> Regarding your concerns with the relevance of the proposed PC-Net to the proposed WSCMR task and whether the components are newly proposed, we have detailed them in the rebuttal, and we believe these details have addressed your concerns.
>
> We understand that reviewers have limited time to provide feedback, but the discussion deadline is approaching. If you have any further questions or concerns, please do not hesitate to let us know so that we can clarify them.
>
> Thank you again for your time and consideration.
>
> Best wishes!

---

### Official Review · Reviewer_SEF7 · 2025-07-05

**Clarity:** 3
**Significance:** 2
**Originality:** 3
**Rating:** 4
**Confidence:** 4

**Summary:**

This paper introduces a new task—Weakly Supervised Compositional Moment Retrieval (WSCMR)—and proposes a novel architecture, PC-Net, to tackle it. The authors target the generalization challenge in moment retrieval, especially under weak supervision and compositional queries with novel syntactic structures or vocabulary. PC-Net incorporates a dual-granularity proposal generator, a semantic-aligned feature aggregator, and a quality margin regularizer. Experiments on Charades-CG, ActivityNet-CG, TACoS and Ego4D demonstrate strong performance and generalization with high parameter efficiency.

**Questions:**

1. The paper claims existing methods poorly generalize to compositional queries with novel syntactic structures or vocabulary, but the 'novel queries' in experiments only use simple combinatorial patterns. Queries with higher semantic richness (e.g., diverse sentences generated by LLMs) should be tested.

2. The model structure appears over-engineered and insufficiently concise. How to validate the rationality of these structures? For example, disassemble the modules into existing frameworks to verify their generalizability.

3. The performance improvement seems marginal despite lower parameters. Could it achieve greater gains at the same parameter level?

**Ethical Concerns:**

["NO or VERY MINOR ethics concerns only"]

**Final Justification:**

After reading the authors' rebuttal, most of my concerns have been well addressed. I am leaning to my current positive ratings.

**Limitations:**

Yes.

**Quality:**

2

**Strengths And Weaknesses:**

Paper Strengths

Novel Task Definition: Clearly motivated definition of WSCMR addressing practical scalability issues.
Methodological Innovation: The integration of dual-granularity proposal generation and semantic-aware aggregation is well-executed and contributes to generalization.

Performance and Efficiency: Achieves state-of-the-art results with significantly fewer parameters than existing baselines.

Thorough Evaluation: Demonstrates effectiveness and generalization across multiple datasets (Charades-CG, ActivityNet-CG, TACoS, Ego4D) with thorough ablations and visualizations.

Clarity and Structure: The paper is well-organized with sufficient technical detail and intuitive figures.

Major Weaknesses

Limited Analysis of Weak vs. Full Supervision: While the model demonstrates strong performance, the paper lacks a deeper analysis of how weak supervision compares to full supervision in terms of interpretability, robustness, and training stability. A more thorough exploration of this trade-off would strengthen the contribution.

---

> ### Author Rebuttal · Authors · 2025-07-31
>
> ## Weaknesses
>
> > 1. **While the model performs well, it lacks analysis of interpretability, robustness, and training stability of fully and weakly supervised methods, which contributes to this paper's contribution.**
>
> - Thank you for your valuable suggestions and recognition of the model's performance. Firstly, the core difference between fully supervised and weakly supervised methods is whether the training phase relies on accurate timestamps of the relevant moments. The reliance of fully supervised methods on cumbersome timestamp annotations leads to their low scalability in new application scenarios.
> - Secondly, we will make the following additions to strengthen the contribution as suggested:
>   - In terms of interpretability, we will add cross-modal attention visualization figures to intuitively compare and show the frame-level query relevance learned by the fully supervised and weakly supervised methods, to help illustrate that the weakly supervised method also has strong semantic learning capabilities.
>   - For robustness, we rewrite the queries in the most representative Charades-CG and ActivityNet-CG Test-trivial sets based on the latest stable version of Qwen3 to simulate different query styles. As shown in Table R1 in Q1, the performance of the proposed PC-Net under the new queries is still better than the existing weakly supervised methods, highlighting its powerful proposal representation capabilities, enabling it to better adapt to the diverse descriptions of real scenes, verifying its robustness advantage (the new query used in the test will be made public in the open source repository).
>   - Finally, to support the training stability of the proposed method, we will supplement the corresponding loss function decline curve.
>
> ## Questions
>
> > 1. **The 'novel queries' in experiments only use simple compositional patterns. Queries with higher semantic richness (e.g., diverse sentences generated by LLMs) should be tested.**
>
> Table R1: Comparison of Test-Trivial split rewriting queries based on the Charades-CG and ActivityNet-CG datasets, using Qwen3 for query rewriting to obtain semantically consistent and diverse queries. 'Fully' means the fully supervised method.
>
> |Model|Rewrite queries using Qwen3|Charades-CG||||ActivityNet-CG||||
> |:-:|:-|:-|:-|:-|:-|:-|:-|:-|:-|
> |||**R1@0.3**|**R1@0.5**|**R1@0.7**|**mIoU**|**R1@0.3**|**R1@0.5**|**R1@0.7**| **mIoU**|
> |QD-DETR$_s$(Fully)|No|-|60.66|38.60|52.53|-|43.76|25.98|42.86|
> |PC-Net(Ours)|No|70.87|54.84|26.68|47.12|52.78|29.62|14.35|36.45|
> |QD-DETR$_s$(Fully)|Yes|65.67|53.62|32.53|46.27|53.18|35.76|20.23|38.04|
> |CNM|Yes|43.86|29.46|14.02|29.49|45.86|25.23|13.43|33.01|
> |CPL|Yes|44.64|31.88|14.50|29.85|45.77|24.00|11.60|31.26|
> |CCR|Yes|59.14|45.06|21.38|39.28|44.65|26.95|10.58|33.92|
> |QMN|Yes|55.94|40.50|18.67|37.08|45.79|24.42|13.14|33.73|
> |PPS|Yes|59.82|43.02|20.93|39.31|48.36|26.13|13.24|31.71|
> |PC-Net(Ours)|Yes|**61.18**|**47.13**|**22.61**|**40.90**|**49.85**|**27.64**|**14.38**|**34.59**|
>
> - Thanks for your suggestions.
>   - To make a fair comparison with existing methods, our experimental settings in the original paper follow the previous study (QD-DETR$_s$), using the same compositional mode (e.g., 'Novel-Composition' and 'Novel-Word') to verify the generalization performance of different methods.
>   - As suggested, to test the robustness of models to diverse query expressions, we have additionally used Qwen3 to perform query rewriting on the Test-trivial set of the most representative Charades-CG and ActivityNet-CG datasets. We will further add these convincing experiments to the original paper to demonstrate the PC-Net's effectiveness.
>
> - As shown in Table R1, our method maintains good performance under this new expression style, fully demonstrating its generalization ability to different query styles. Even under the more challenging ActivityNet-CG mIoU metric, PC-Net experiences less performance degradation (-1.86%) than the fully supervised method QD-DETR$_s$ (-4.82%) when faced with queries with widely varying expression styles. This is due to the adaptation of the constructed dual-granularity proposal generator to potentially complex temporal logic queries, the good proposal boundaries obtained, and the powerful proposal representation ability of the aligned proposal feature aggregator, which makes the discriminative proposal easy to select. In subsequent work, we will further explore more complex forms of semantic combination.
> - The LLM prompts and examples for rewriting queries are shown below:
>
> ```python
> messages=[
>             {
>                 "role": "system",
>                 "content": "You are a video content retrieval assistant specialized in query rewriting. "
>                            "Your task is to generate semantically identical but linguistically diverse versions "
>                            "of video-related queries while strictly preserving their original meaning."
>             },
>             {
>                 "role": "user",
>                 "content": f"Rewrite the following video moment retrieval query into a semantically equivalent "
>                            f"but linguistically diverse version. Requirements:\n"
>                            f"1. Preserve the exact meaning of the original query.\n"
>                            f"2. Alter phrasing (e.g., syntax, vocabulary, active/passive voice).\n"
>                            f"3. Ensure compatibility with video content retrieval (e.g., action/object/scene descriptions).\n\n"
>                            f"Original Query: \"{original_query}\"\n\n"
>                            f"Rewritten Query:"
>             }
>         ],
> ```
>
> | Original                        | Rewritten                                                |
> | ------------------------------- | -------------------------------------------------------- |
> | person pouring it into a glass. | An individual transferring liquid into a drinking glass. |
> | person put the clothes.         | Someone places the garments.                             |
>
> > 2. **The model structure appears over-engineered and insufficiently concise. How to validate the rationality of these structures? For example, disassemble the modules into existing frameworks to verify their generalizability.**
>
> - We clarify that the proposed PC-Net is not over-designed, and its process is relatively clear, as recognized by reviewers #H739. As shown in the model overview in Figure 2, the key modules contain three parts around the proposal, which are boundary generation, feature modeling, and quality assessment. The components are designed based on the two challenges faced by WSCMR: **(1)** weak pairwise supervision and limited query style in training, and **(2)** the need to generalize to compositional queries with diverse temporal and semantic dependencies.
> - We are pleased that the reviewer raised the issue of component rationality, which is also a key consideration in model construction. In fact, we conducted detailed component combination experiments in the ablation experiments to verify the contribution of each component to generalization ability through rigorous ablation experiments based on the existing framework CPL (Setting a) , as shown in Table 3:
>   - The dual-granularity proposal generator and cross-modal semantic contrastive loss significantly alleviate the problem of insufficient fine-grained temporal perception and cross-modal gap (compare rows (b)&(d) and Ours);
>   - The learnable peak-aware Gaussian distributor enhances proposal features by modeling action duration diversity (row (j) & Ours);
>   - The quality margin regularizer effectively enhances visual-semantic association learning and suppresses the potential false correlation (row (h) & Ours).
> - In summary, the innovation of PC-Net lies in alleviating the generalization challenge of weakly supervised scenarios: dual-granularity proposal generator handles complex temporal dependencies, proposal feature aggregator narrows the semantic gap and adapts to variable action duration, and the quality margin regularizer suppresses spurious associations. These components work closely together to deal with the core challenge of WSCMR.
>
> > 3. **The performance improvement seems marginal despite lower parameters. Could it achieve greater gains at the same parameter level?**
>
> - Thank you for your insightful questions. The performance gains are not marginal, given the efficiency and inherent difficulty of the task. First, as shown in Table 1, our method achieves a 0.58% mIoU gain on the Novel-Composition split and a 1.86% mIoU gain on the Novel-Word split compared to CPL, while only increasing the number of parameters by 0.33M. Crucially, it also surpasses the fully supervised VISA (CVPR'22) method on the Test-Trivial split. In contrast, the CCR method uses three times as many parameters as CPL but performs inferior to CPL, highlighting the challenges of WSCMR.
>
> - Our core contribution is a concise and reproducible framework designed to address the unique challenges of WSCMR, such as compositional generalization and weak paired supervision. A key design principle is parameter efficiency, achieving competitive gains with minimal additional complexity. As mentioned in the conclusion, future work will explore enhancing the understanding of compositional queries in long videos, possibly through modules such as refined temporal modeling, which could increase the number of parameters and further improve performance.

---

> ### Author Response · Authors · 2025-08-05
>
> Dear reviewer #SEF7:
>
> Thank you for your time and careful review.
>
> About your concerns regarding the in-depth analysis of weakly and fully supervised methods, we have added robustness experiments and analysis. We also intend to visualize interpretability and training stability as suggested to highlight the contributions of this paper.
>
> In addition, we have added details on the robustness experiments, an analysis of the rationality of the proposed components, and a discussion of parameter sizes. We believe these details can address your concerns.
>
> We note that you have already submitted your final score, and thank you for your proactive action. If you have any further questions or concerns, please let us know so we can clarify.
>
> Thank you again for your valuable time and suggestions.
>
> All the best!

---

### Decision · Program_Chairs · 2025-09-17

**Decision:**

Accept (poster)

**Comment:**

This paper receives positive scores. The reviewers raise some concerns regarding the limited analysis of weak vs. full supervision, the limited motivation for each component design, the complex model architecture, the marginal performance improvement, etc. Most of the questions are addressed in the rebuttal and recognized by reviewers. Eventually, this paper is recommended to be accepted. Authors should incorporate these review feedbacks, and polish the paper writing in the final version.